# Geminivirus-encoded TrAP suppressor inhibits the histone methyltransferase SUVH4/KYP to counter host defense

Claudia Castillo-González[1,2], Xiuying Liu[1,2,3], Changjun Huang[1,2,4,5], Changjiang Zhao[1,2,6], Zeyang Ma[1,2], Tao Hu[1,2,4], Feng Sun[1,2,7], Yijun Zhou[7], Xueping Zhou[8], Xiu-Jie Wang[3], Xiuren Zhang[1,2]*

[1]Department of Biochemistry and Biophysics, Texas A&M University, College Station, United States; [2]Institute for Plant Genomics and Biotechnology, Texas A&M University, College Station, United States; [3]State Key Laboratory of Plant Genomics, Institute of Genetics and Developmental Biology, Chinese Academy of Sciences, Beijing, China; [4]State Key Laboratory of Rice Biology, Institute of Biotechnology, Zhejiang University, Hangzhou, China; [5]Yunnan Academy of Tobacco Agricultural Sciences, Yunnan, China; [6]College of Agronomy, Heilongjiang Bayi Agricultural University, Daqing, China; [7]Institute of Plant Protection, Jiangsu Academy of Agricultural Sciences, Nanjing, China; [8]Biotechnology Institute, College of Agriculture & Biotechnology, Zhejiang University, Zhejiang, China

*For correspondence: xiuren.zhang@tamu.edu

Competing interests: The authors declare that no competing interests exist.

**Abstract** Transcriptional gene silencing (TGS) can serve as an innate immunity against invading DNA viruses throughout Eukaryotes. Geminivirus code for TrAP protein to suppress the TGS pathway. Here, we identified an *Arabidopsis* H3K9me2 histone methyltransferase, Su(var)3-9 homolog 4/Kryptonite (SUVH4/KYP), as a bona fide cellular target of TrAP. TrAP interacts with the catalytic domain of KYP and inhibits its activity in vitro. TrAP elicits developmental anomalies phenocopying several TGS mutants, reduces the repressive H3K9me2 mark and CHH DNA methylation, and reactivates numerous endogenous KYP-repressed loci in vivo. Moreover, KYP binds to the viral chromatin and controls its methylation to combat virus infection. Notably, *kyp* mutants support systemic infection of TrAP-deficient Geminivirus. We conclude that TrAP attenuates the TGS of the viral chromatin by inhibiting KYP activity to evade host surveillance. These findings provide new insight on the molecular arms race between host antiviral defense and virus counter defense at an epigenetic level.

## Introduction

RNA silencing is a host defense mechanism to combat invading nucleic acids. One type of RNA silencing is referred to as post-transcriptional gene silencing (PTGS). In PTGS, double-stranded RNAs (dsRNAs) are processed by Dicer-like ribonucleases into small-interfering RNAs (siRNAs). Mature siRNAs are incorporated into an Argonaute (AGO)-centered RNA-induced silencing complex (RISC) to regulate expression of target genes through RNA cleavage or translational repression. PTGS has evolved as a universal defense response toward all viruses because dsRNAs can result from intermediates in RNA virus replication, highly structured RNA virus genomes, or from viral transcripts. To evade this surveillance mechanism, virtually all plant viruses are known to encode suppressor proteins that are able to block different key steps of the PTGS pathway (*Ding and Voinnet, 2007*).

While the host/virus battle at the PTGS level has been well appreciated, virus suppression at a transcriptional gene silencing (TGS) level is poorly understood. In eukaryotes, the nuclear DNA is

**eLife digest** Many viruses can infect plants and cause diseases that can reduce crop yields. The Geminiviruses are a family of plant viruses that are transmitted by insects and infect tomato, cabbage, and many other crop plants. These viruses hijack the plant cells that they infect and force the plant cells to make viral proteins using instructions provided by the genes in the virus' own DNA. To make proteins, DNA is first copied into molecules of messenger ribonucleic acid (or mRNA) in a process called transcription. However, plants can defend themselves by blocking the transcription of viral DNA through 'transcriptional gene silencing'.

In plant cells, DNA is packaged around proteins called histones to form a structure called chromatin. Small chemical tags attached to the histones can alter the structure of chromatin to regulate the activity of the genes encoded within it. For example, 'methyl' tags added to certain histones can block transcription and lower the activity of a gene. DNA from viruses can also associate with histones inside plant cells meaning that transcriptional gene silencing can take place by the addition of these methyl tags. Many Geminiviruses produce a protein called TrAP, which can activate transcription, but it is not clear how this works.

Castillo-González et al. studied the TrAP proteins from two different Geminiviruses that can infect crop plants. The commonly used model plant, *Arabidopsis thaliana*, was genetically engineered to produce high levels of these TrAP proteins. These 'transgenic' plants did not develop properly: they grew more slowly, had abnormal leaves, and flowered earlier. Furthermore, hundreds of plant genes were more active than usual in the transgenic plants, which suggests that TrAP inhibits transcriptional gene silencing. Further experiments showed that TrAP directly binds to a plant enzyme called KYP—which normally deposits methyl groups on chromatin and prevents it from working. This reduces the number of methyl groups that are attached to histones associated with both viral and plant chromatin, which results in the activation of genes that would normally be switched off.

Castillo-González et al.'s findings show how Geminiviruses can stop transcriptional gene silencing of chromatin that contains virus DNA to evade the host plant's defenses. The next challenge is to understand how TrAP inhibits KYP, which may present new ways to genetically engineer plants to become resistant to infection by viruses.

wrapped onto histone octamers to constitute chromatin. The chromatin undergoes various DNA and histone methylations, and these modifications have variable effects on gene expression depending on the precise residues, contexts, and modification complexity. Histone methylation takes place on lysine and arginine residues of the amino-terminal tails (*Kouzarides, 2007*; *Greer and Shi, 2012*). The prevailing dogma is that histone 3 lysine 4 tri-methylation (H3K4me3) is mostly associated with transcriptionally active euchromatin, while H3K9me2 and H3K27me3 are repressive marks (*Deal and Henikoff, 2011*; *Feng and Jacobsen, 2011*). Histone methylation is catalyzed by SET domain containing methyltransferases, specifically, H3K9me2 is deposited by *Arabidopsis* Su(var)3-9 homolog 4, Kryptonite (KYP) (*Du et al., 2014a*), and its paralogs (SUVH5,6), while H3K27 methylation is carried out by the Polycomb repressive complex 2 (PRC2), which includes Curly Leaf (CLF) (*Liu et al., 2010*; *Zheng and Chen, 2011*). Local H3K9me2 and H3K27me3 can spread over wide regions to elicit heterochromatin configuration. In animals, the propagation of histone methylation entails co-repressor heterochromatin protein 1 (HP1), whereas in plants, KYP acts synergistically with DNA methyltransferases (i.e., Chromomethylase 3 [CMT3]) to constitute a mutually reinforcing cycle of DNA and histone methylation to secure TGS (*Du et al., 2012*, *2014a*).

Histone methylation not only regulates endogenous gene expression but also invasive DNAs such as transposons and viruses (*Narasipura et al., 2014*). Plant DNA viruses, exemplified by Geminivirus, form minichromosomes in the host (*Hanley-Bowdoin et al., 2013*). Both Geminivirus DNA and associated histones are methylated in infected cells, whereas viral methylation is reduced in methylation-deficient hosts, methylation-compromised *Arabidopsis* mutants are hypersusceptible to Geminivirus infection and show exacerbated disease symptoms (*Raja et al., 2008*). Thus, plants appear to employ methylation of viral chromatin to limit viral replication and transcription (*Aregger et al., 2012*; *Pumplin and Voinnet, 2013*). On the other hand, Geminiviruses encode a

multi-functional protein called transcriptional activation protein (TrAP/AL2/AC2) that counters the epigenetic defense (*Raja et al., 2008*; *Buchmann et al., 2009*). It has been shown that TrAP inhibits adenosine kinase (ADK) (*Wang et al., 2005*). ADK catalyzes the synthesis of 5′ AMP from adenosine and ATP, a process that promotes the regeneration of S-adenosyl-methionine (SAM), the major methyl donor in the cell (*Moffatt et al., 2002*; *Buchmann et al., 2009*). Consequently, the TrAP-mediated inhibition of ADK activity likely impedes downstream trans-methylation events, including viral chromatin methylation in the nucleus (*Bisaro, 2006*; *Buchmann et al., 2009*). In parallel, some Geminivirus encode a TrAP positional homolog, named C2, that is able to stabilize SAM decarboxylase 1 to downregulate the methyl group metabolism (*Zhang et al., 2011*). It seems that interfering with the methyl cycle is a common suppression mechanism for Geminivirus-encoded TrAP/AL2/C2 proteins. In addition, C2 also subverts the activity of COP9 signalosome to inhibit jasmonate signaling (*Lozano-Durán et al., 2011*), suggesting its multiple functions in viral counter-defense.

Here, we investigated the suppression mechanism of TrAP proteins, encoded by two Geminivirus members, *Tomato Golden Mosaic Virus* (TGMV) and *Cabbage Leaf Curl Virus* (CaLCuV). We found that constitutive expression of TGMV-*TrAP* in *Arabidopsis thaliana* caused morphological abnormalities that mimic loss-of-function mutants of numerous TGS components including *lhp1* (*like-heterochromatin1*) and *clf*. Microarray analyses of *TrAP* transgenic plants and *lhp1* mutants revealed a substantial overlap in reprogrammed host genes at a genome-wide level. Through biochemical screening, we identified KYP as the bona fide target of TrAP. We demonstrated in vitro that TrAP binds to the catalytic domain of KYP and inhibits its enzymatic activity; while in vivo, *TrAP* decreases the repressive H3K9me2 marks and H3K9me2-dependent CHH methylation in gene-rich regions. We also found that KYP directly associates with the Geminivirus minichromosome and deposits H3K9me2 marks on viral chromatin. In addition, *kyp* mutants but not wild-type plants sustain low systemic infection of CaLCuV lacking TrAP protein. Taken together, we propose that KYP-catalyzed H3K9me2 is a line of the innate immunity against invading DNA pathogens, and Geminivirus TrAP functions to inactivate KYP to counter host defense. Thus, this study provides new insight into the host–virus interaction at the TGS level.

## Results

### TGMV-encoded TrAP causes developmental abnormalities in *Arabidopsis* but not through miRNA pathway

To study the suppression mechanism of TrAP, we generated 235 *Arabidopsis* transgenic lines overexpressing full-length TGMV TrAP, with or without Flag-Myc4, 3HA, or CFP epitopes. These transgenic plants were confirmed by northern (data not shown) or western blot assays (*Figure 1A* and *Figure 3—figure supplement 1A*). Importantly, the majority of the transgenic lines exhibited developmental abnormalities consisting of short statues, strongly upward curled cotyledons and true leaves (*Figure 1B*). Moreover, these overexpressing lines exhibited early flowering compared to wild-type (WT) plants. These phenotypes were morphologically distinct from loss-of-function mutants of *ADK1 ADK2* (*Weretilnyk et al., 2001*; *Moffatt et al., 2002*), *SnRK1* (*Shen et al., 2009*, *2014*), *PEAPOD2* (*Lacatus and Sunter, 2009*), and *rgsCaM* (*Chung et al., 2014*), a calmodulin-like protein, which are also targets or partners of TrAP. This result indicated that TrAP exerts some novel cellular function (s).

Developmental anomalies of transgenic plants expressing viral suppressors have been associated with interruption of the miRNA pathway. To test this, we compared expression levels of several miRNAs and their targets between Col-0 wild-type and *TGMV TrAP* transgenic plants. Plants expressing *Cucumber mosaic virus*-encoded 2b suppressor (Zhang et al., 2006) and *ago1-27*, a hypomorphic allele of *ago1* (*Morel et al., 2002*), were used as controls. We observed that the accumulation of miR165 and miR168 and their targets, *PHB* and *AGO1* transcripts, in the *35S-TGMV TrAP* lines was comparable to the amount in wild-type plants (*Figure 1C,D*). We further confirmed that loading of miRNAs into AGO1-centered RISCs was not affected by TrAP (*Figure 1E*). The same results were obtained with miR167, miR159, and ta-siRNA480/255 and their corresponding targets (*Figure 1—figure supplement 1*). Thus, unlike most of previously reported viral suppressors, TrAP does not act on the miRNA pathway.

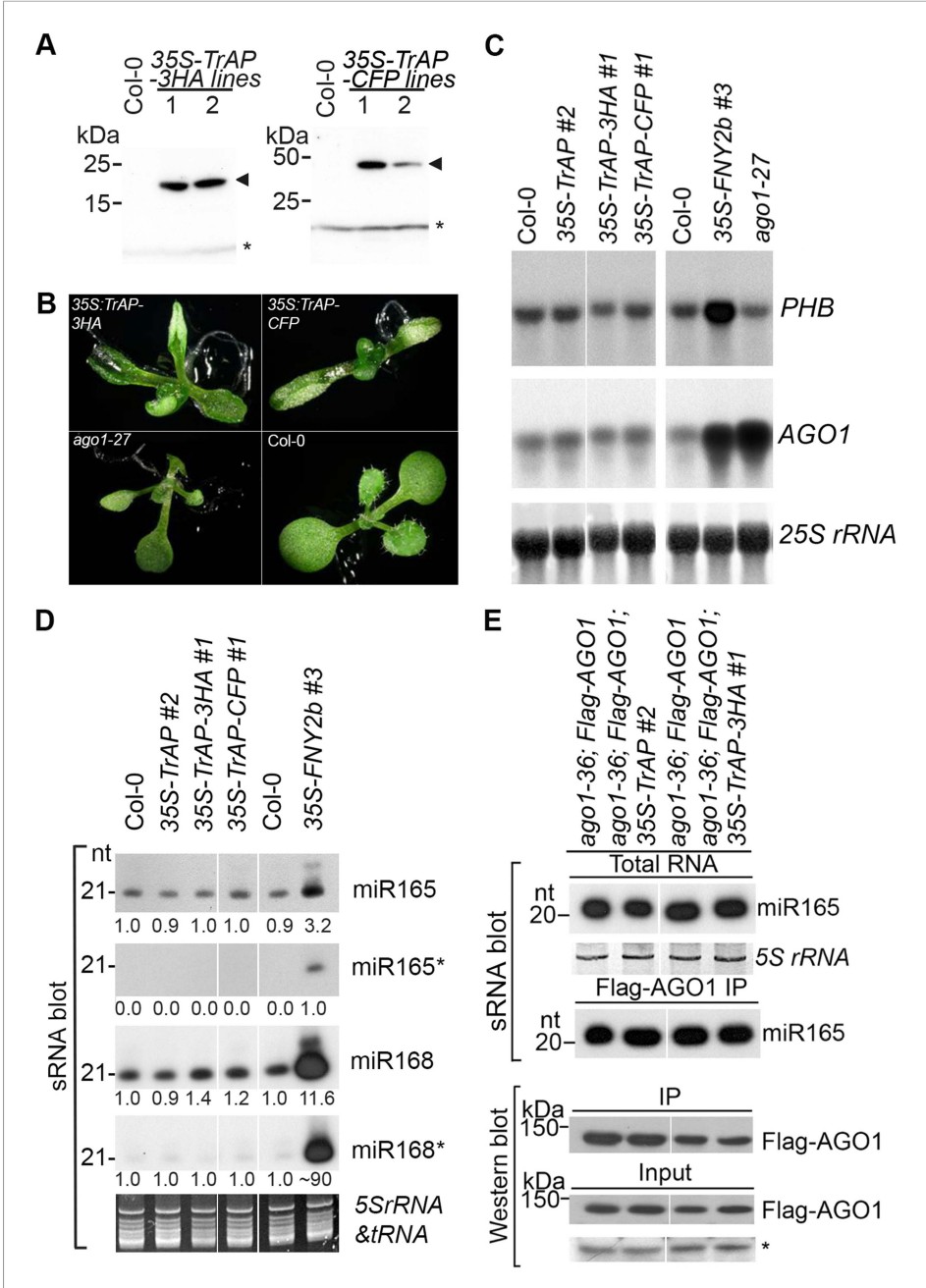

**Figure 1**. TrAP caused developmental abnormalities in Arabidopsis but not through miRNA pathway. (**A**) Western blot analysis of *35S-TrAP-3HA* and *35S-TrAP-CFP* transgenic lines. Arrows indicate the locations of the tagged TrAP proteins; * Cross-reaction band serves a loading control. (**B**) Morphological defects of Arabidopsis transgenic plants expressing *35S-TGMV TrAP*. Photographs were taken of 10-day seedlings. (**C**) RNA blot analysis of *PHB* and *AGO1* transcripts in the *TrAP* overexpression transgenic plants using gene-specific random-labeled probes. *25S rRNA* is a loading control. (**D**) sRNA blot analysis of miRNA and miRNA* in the *TrAP* overexpression transgenic plants. Total RNA was prepared from a pool of T2 transgenic plants (n > 50 for each line). sRNA blots were probed using 5′ end $^{32}$P-labeled oligonucleotide probes complementary to the indicated miRNA or miRNA*. *5S rRNA* and *tRNA* are a loading control. All the samples were run in the same gel; but the lane order of miRNA*s was rearranged to match that of miRNAs. (**E**) sRNA blot analysis of miR165 loading into Arabidopsis RNA-induced silencing complexes (RISCs). RNA was extracted from flowers or Flag-AGO1 immunoprecipitates of transgenic plants harboring *35S-TrAP* or *35S-TrAP-3HA* in *ago1-36; P$_{AGO1}$-Flag-AGO1* background and *ago1-36; P$_{AGO1}$-Flag-AGO1* control plants (**Baumberger and Baulcombe, 2005**). Top panel, total RNA; middle panel, each lane contained sRNA associated

*Figure 1. continued on next page*

*Figure 1. Continued*

with Flag-AGO1 immunoprecipitated from 0.4 g of flowers. Bottom panel, the input and immunoprecipitate of Flag-AGO1 were analyzed by Western blot assays in the same samples for sRNA blots. A cross-reacting band (*) was used as a loading control.

The following figure supplement is available for figure 1:

**Figure supplement 1**. sRNA blot analysis of additional miRNAs and siRNA in the TrAP overexpression transgenic plants.

## TrAP genetically interrupts the TGS pathway

To study how *TrAP* altered plant development, we mined publicly available databases and literature for the molecular and morphological phenotype of *35S-TrAP* lines. We found that *35S-TrAP* transgenic lines phenocopied several mutants in the epigenetic pathway including *lhp1* (*Kotake et al., 2003*) (*Figure 2A*) and *clf* mutants (*Chanvivattana et al., 2004*), with respect to the early flowering and upward curling of leaves. CLF belongs to PRC2, a complex that catalyzes the deposition of H3K27me3 marks. LHP1 (*Nakahigashi et al., 2005*), on the other hand, associates to silent genes in euchromatin and directs the spreading of the silent status to adjacent loci (*Turck et al., 2007*; *Zhang et al., 2007a, 2007b*; *Farrona et al., 2008*; *Zheng and Chen, 2011*). Thus, coordinate activities of CLF and LHP1 result in chromatin methylation and transcriptional repression (*Farrona et al., 2008*).

We examined global expression profiles of 7-day-old *35S-TrAP* transgenic plants compared to Col-0 wild-type using an Affymetrix ATH1 GeneChip and identified 586 genes that are differentially expressed in the *35S-TrAP* transgenic plants (q < 0.005). Of these, 295 transcripts were elevated whereas 291 were reduced (*Figure 2E*). We performed real-time PCR and RNA blot assays to validate the microarray results for the differentially expressed genes (DEGs). Among 25 genes randomly tested, we confirmed the ATH1 results for 24, indicating that the microarray results were reliable (*Figure 2B,C*; data not shown). Gene-Ontology (GO) analysis placed the DEGs into seven functional categories (*Figure 2D*; *Supplementary file 1*): hormone response (86 genes), stress response (94 genes), development regulation (50 genes), transcriptional regulation (20 genes), RNA metabolism (13 genes), post-translational modification (36 genes), and general metabolism (182 genes), plus a set of 105 un-annotated genes (*Figure 2D*; *Supplementary file 1*).

Then, we compared the DEG profiles of the *TrAP* overexpression lines and loss-of-function *lhp1* mutant (*Figure 2E,F*; *Supplementary file 2*). Transcriptome analysis revealed that out of 295 genes significantly upregulated in the *35S-TrAP* transgenic lines, 120 (40.7%) were also upregulated in *lhp1* mutant. This is significantly greater than 1.28% expected by chance (p < 2.2e$^{-16}$, Pearson's Chi-squared test). Interestingly, among the co-upregulated genes are a group of flowering-stimulated transcriptional factors including the key flowering-time integrator, *FT,* and 12 other genes such as *TFL1*, *AGL5*, and *AGL9* (*Farrona et al., 2008*) (*Figure 2F–H*; *Supplementary file 2*; *Supplementary file 8*) clustered in the transcriptional regulation category. Importantly, all these genes are regulated through epigenetic pathways and account for the early flowering phenotypes of *lhp1* mutant and possibly of *TrAP* transgenic plants as well (*Gan et al., 2013*). Other highly represented categories included 31 genes involved in aging and 116 genes engaged in stress responses (*Supplementary file 2*). Notably, the stress-responsive genes included genes specific to biotic stress such as *PR4*, *WRKY18*, *FLS2*, and *PDF1.2*; additionally, genes related to chemical stress, such as *PTR3* and *TAT3*, were also identified. Thus, constitutive expression of *TrAP* could trigger plant senescence and innate defense pathways, and this activation is potentially through interference with the *LHP1*-related epigenetic silencing.

Similarly, out of 291 genes significantly downregulated in the *35S-TrAP* transgenic lines, 137 (47.1%) were also repressed in *lhp1* mutant. This is significantly greater than 1.25% expected by chance (p < 2.2e$^{-16}$, Pearson's Chi-squared test) (*Supplementary file 2*). Genes related to auxin response were of special interest. Of the 32 DEGs involved in the auxin pathway, 29 genes were downregulated in the *35S-TrAP* transgenic plants, classified as small auxin upregulated mRNAs (*SAURs*). These results suggested a possible hyposensitivity to auxin in *lhp1* mutants and *TrAP* transgenic plants, which could explain the smaller statues of both genotypes. Consistent with this

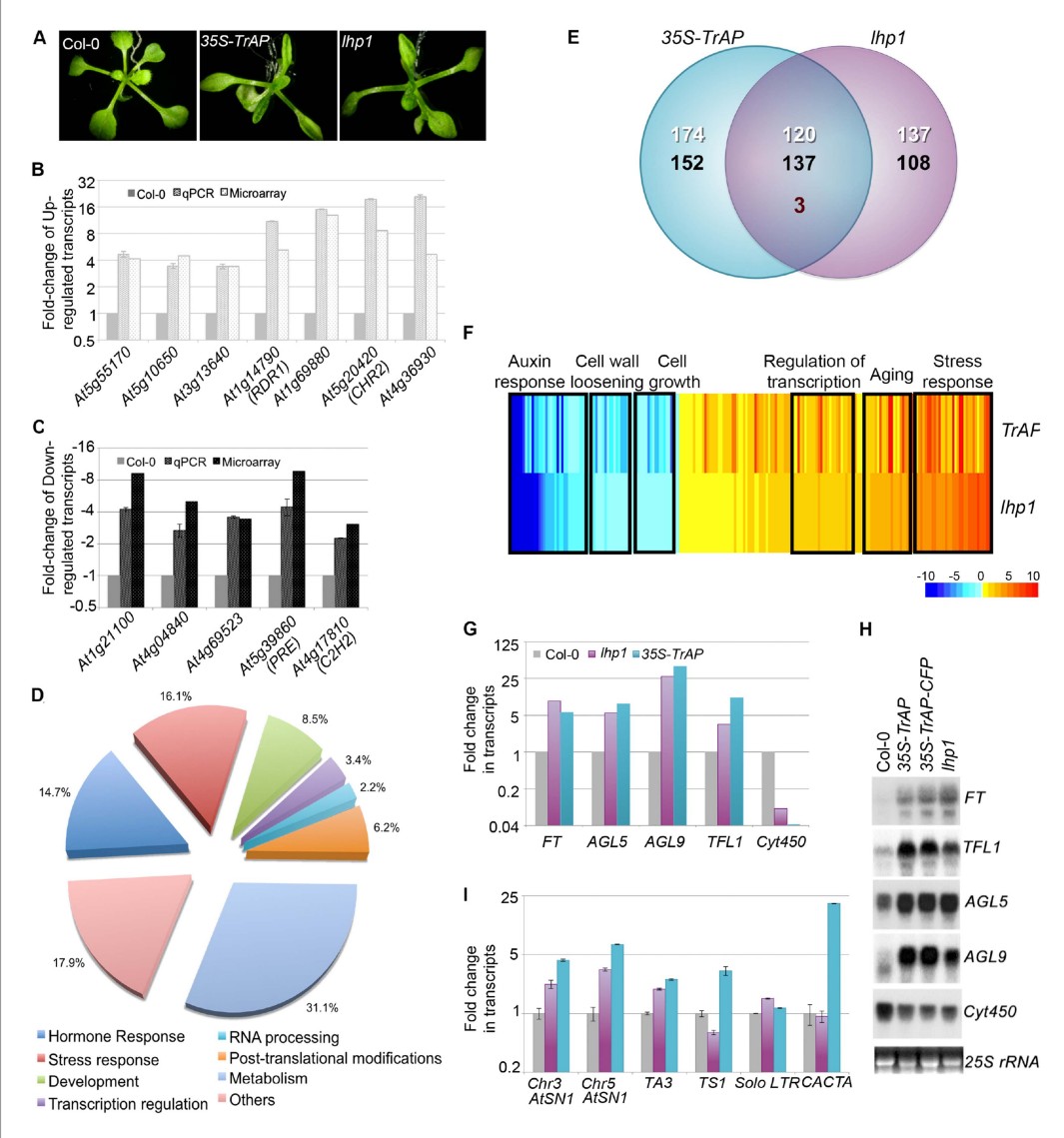

**Figure 2**. TrAP is genetically involved in the TGS pathway. (**A**) *35S-TrAP* transgenic plants phenocopied *lhp1* mutants. Photographs were taken of 15-day seedlings. (**B, C**) Microarray results were validated by qRT-PCR analysis. Only 12 randomly selected loci were shown. (**D**) Gene ontology analysis of the *TrAP*-regulated DEGs. The numbers adjacent to the pies represent the ratio of genes in each category over the total DEGs. (**E**) Genome-wide overlapping of the genes regulated by *TrAP* and loss-of-function *lhp1*. White and black numbers correspond to upregulated and downregulated genes, respectively. Maroon number indicates the genes that are differentially deregulated in both genotypes. (**F**) Heatmap of the commonly deregulated genes in the *35S-TrAP* and *lhp1* lines. The typical gene-ontology categories are shown on top. (**G**, **H**) Microarray and RNA blot analyses of epigenetically regulated flowering genes in the *TrAP* transgenic lines and *lhp1* mutants. *Cyt450* is a control. (**I**) qRT-PCR analysis of TEs in heterochromatic regions in the *lhp1* mutant and *TrAP* transgenic lines.

hypothesis, numerous auxin-repressed loci including *PS2* and aging genes like *TET9*, *SAG13*, and *SRG1* were upregulated in both lines. Concomitantly, six genes related to cell growth and five genes engaged in cell wall loosening were also repressed. Significant genome-wide overlap of *35S-TrAP* and *lhp1* loss-of-function-responsive genes suggested that *TrAP* might be genetically involved in the *LHP1*-related TGS pathway.

Since LHP1 is believed to reside in euchromatic regions, we wondered whether *TrAP* also deregulates expression of heterochromatic loci. To this end, we selected numerous transposable

elements (TEs) that were not recovered from the microarray assays and assessed them directly by qRT-PCR. Excitingly, most of the tested transposons were transcriptionally active (*Figure 2I*), further suggesting that *TrAP* indeed inhibits the TGS pathway, in both euchromatic and heterochromatic regions.

## KYP is a bona fide target of TrAP

Given that *TrAP* transgenic plants phenocopied several TGS mutants and displayed transcriptional activation of heterochromatic loci, we hypothesized that TrAP epistatically regulates a TGS integrator (s), indirectly leading to deregulation of the epigenetic marks. Analysis of the microarray data challenged this possibility as no significant changes in the transcripts of any canonical TGS components were revealed (*Supplementary file 3*). An alternative hypothesis was that TrAP directly interferes with the function of a TGS component (s). To test this in an unbiased manner, we used luciferase complementation imaging (LCI) assay to screen 34 TGS-related proteins and some other cellular factors for TrAP interaction (*Zhang et al., 2011*) (*Supplementary file 4*). In the LCI experiments, the N- and C-terminal parts of firefly luciferase (NLuc and CLuc) are fused to different test proteins to be transiently expressed in *Nicotiana benthamiana* (*N. benthamiana*). When NLuc and CLuc are brought together through interaction of the test proteins, catalytic activity is restored and recorded through CCD camera. In our LCI screening, we recovered LHP1 and KYP, a SUVH-type H3K9me2 methyltransferase, suggesting that TrAP is physically close to LHP1 and/or KYP proteins in vivo (*Figure 3B*). Next, we carried out confocal microscopy imaging assays. TrAP co-localized with both LHP1 and KYP in scattered but not yet clearly defined nuclear foci, whereas co-expression of TrAP-CFP and other YFP-tagged proteins in *N. benthamiana* cells did not show such patterns (*Figure 3C*). These observations further suggested that TrAP was in the same complexes as LHP1 or KYP. To further examine if TrAP interacted with these proteins, we conducted co-immunoprecipitation (Co-IP) assays (*Figure 3D*). Interestingly, we validated the TrAP-KYP interaction (*Figure 3D*) but not TrAP-LHP1 (data not shown), indicating that the LCI signal resulting from the TrAP-LHP1 combination likely involved additional cofactors between TrAP and LHP1 in vivo (*Figure 3B*). Alternatively, TrAP-LHP1 interaction might be transient or unstable in our stringent co-IP condition. We observed that expression of CLuc-HA3-KYP in *N. benthamiana* yielded truncated proteins of various lengths that accumulated to comparable levels as the full-length protein; only the full-length KYP showed specific interaction with TrAP, implying the KYP C-terminal domain as the interaction interface with TrAP (*Figure 3D*). We further confirmed the in vivo TrAP-KYP interaction by Förster resonance energy transfer-acceptor bleaching (FRET-AB), using TrAP-CFP as a donor and YFP-KYP as an acceptor. Shortly, FRET involves the energy transfer from an excited donor to an adjacent acceptor when the fluorophores are less than 10 nm apart. If the fluorophores are coupled, the excited donor leads to acceptor emission; bleaching the acceptor allows the emission of the donor to be measured (*Figure 3E*). Consecutive cycles of YFP-KYP bleach/recovery correlated with release/quench on the TrAP-CFP signal (*Figure 3E,F*); the positive FRET-AB (CaLCuV-TrAP 2.699 $\pm$ 0.23, TGMV-TrAP 3.8228 $\pm$ 0.58, $p < 0.05$) corroborated TrAP-CFP/YFP-KYP interaction (*Figure 3G*). To examine whether TrAP interacted with KYP under physiological conditions, we isolated TrAP complexes through two-step immunoprecipitation (*Zhu et al., 2013*) from stable transgenic plants expressing *Flag-Myc4-TrAP* under the inducible promoter (*XVE*) (*Zuo et al., 2000*) followed by mass spectrometry analysis. A total of 624 peptides representing 288 unique sequences were recovered from the TrAP sample; of those, 31 unique peptides matched specifically to KYP/SUVH4 and were not found in control immunoprecipitates using Col-0 plants (*Figure 3—figure supplement 1*). Together, all these assays clearly indicated that TrAP and KYP interact in vivo.

## TrAP binds to the catalytic domain of KYP

We performed in vitro pull-down assays to examine whether TrAP interacts directly with KYP. We found that maltose-binding protein (MBP)-KYP, but not MBP and other MBP-tagged control proteins, was able to pull down GST-tagged TGMV- and CaLCuV-encoded TrAP proteins (*Figure 4A,B*). This interaction was specific as MBP-KYP was unable to pull down GST protein alone. Notably, we did not observe direct LHP1-TrAP interaction in the parallel pull-down experiments (*Figure 4—figure supplement 1*); and this result was consistent with the in vivo Co-IP experiments. From N- to C-termini, KYP contains SRA, PreSET, SET, and PostSET domains. To further define the protein domain

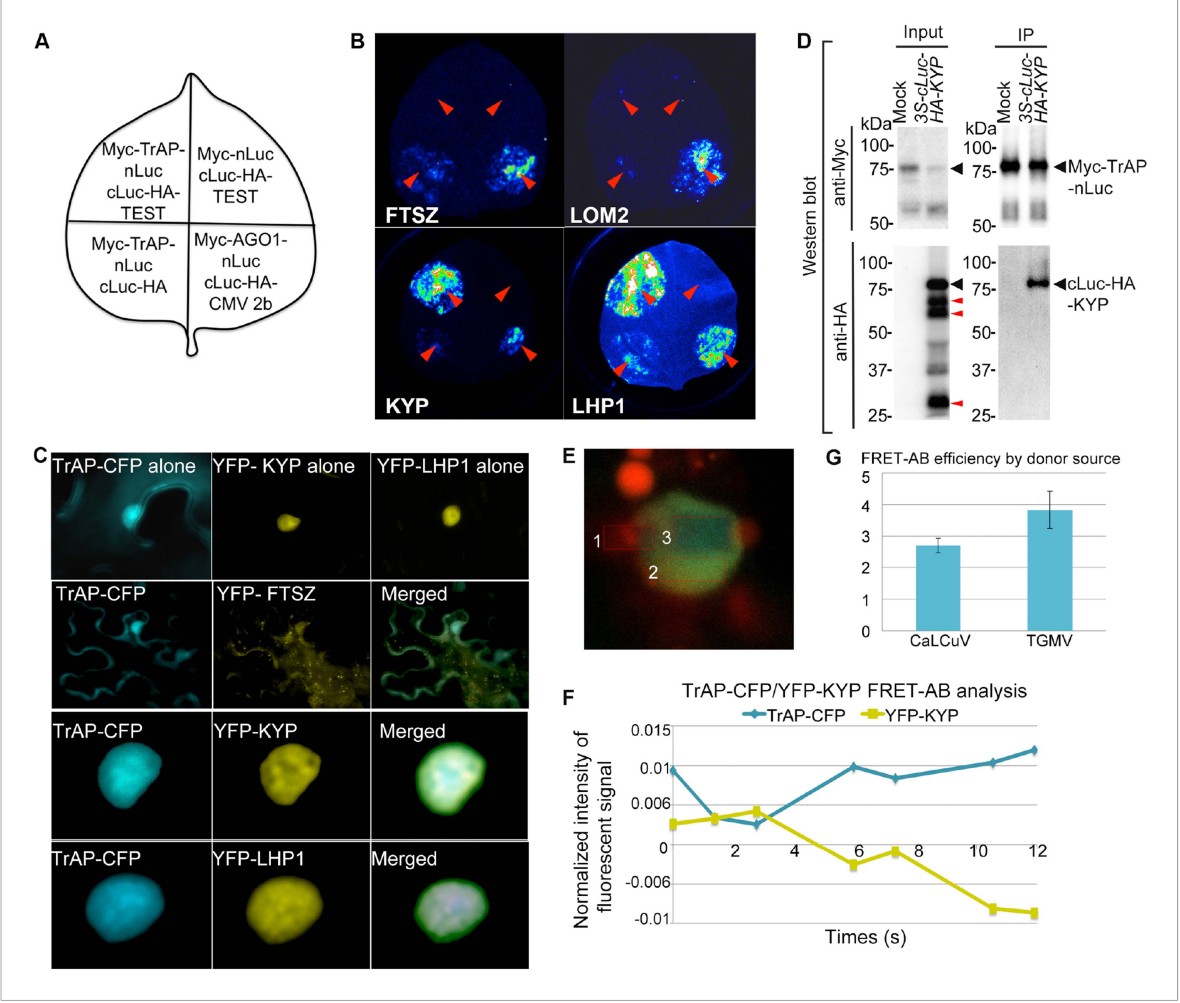

**Figure 3**. TrAP interacts with KYP in vivo. (**A**) Schematic representation of the luciferase complementation imaging assay shows the different combinations of infiltrated constructs that were fused either to N-terminal (NLuc) and C-terminal (CLuc) regions of luciferase. (**B**) Screening of host factors targeted by TrAP. The infiltration positions of the constructs (red arrows) and luminescence signal resulting from the protein–protein interaction in a leaf are shown. FTSZ and LOM2 serve as negative controls. (**C**) Confocal imaging assays show the co-localization of TrAP-CFP with YFP-KYP and YFP-LHP1 in the nuclei in *Nicotiana benthamiana*. FTSZ serves as a negative control. (**D**) Specific interaction between KYP and TrAP was confirmed in *N. benthamiana* by co-immunoprecipitation (Co-IP). Constructs harboring *35S-Myc-TrAP-nLuc* and *cLuc-HA-KYP* were co-infiltrated in *N. benthamiana* leaves. IP was conducted using anti-Myc antibody. Western blot analyses were done with the crude extract (input) and the IP products using anti-Myc or -HA antibodies. Truncated versions (red arrows) serve as an internal control. (**E**) Exemplary imaging of FRET assays of TrAP-CFP and YFP-KYP co-expressed in a nucleus. The nucleus is irradiated with 458 nm laser to excite the CFP fluorophore. Three regions were selected for the assay: #1, autofluorescence control, #2 fluorophore decay control, and #3, FRET-acceptor bleaching test. Regions #1 and #3 were treated with pulses of 514 nm laser to bleach the YFP fluorophore. The CFP signal is then visible in region #3 when the emission of CFP is dequenched. (**F**) Quantification of the signals from each fluorophore observed during FRET-AB experiment in **E**. (**G**) FRET is positive for YFP-KYP paired with the CFP-tagged TrAPs from either CaLCuV or TGMV.

The following figure supplement is available for figure 3:

**Figure supplement 1**. Mass spectrometry analyses confirmed endogenous KYP as a bona fide TrAP-interacting partner.

(s) responsible for the specific interaction, we generated five truncations of KYP (*Figure 4C*). Pull-down assays showed that the SET domain interacted with TrAP proteins from both TGMV and CaLCuV at a higher affinity, whereas SRA domain might interact with CaLCuV TrAP at a reduced affinity in vitro (*Figure 4D–F*). These results were consistent with the results of the in vivo Co-IP experiments in which only full-length KYP, but not C-terminal-truncated versions, could recover TrAP

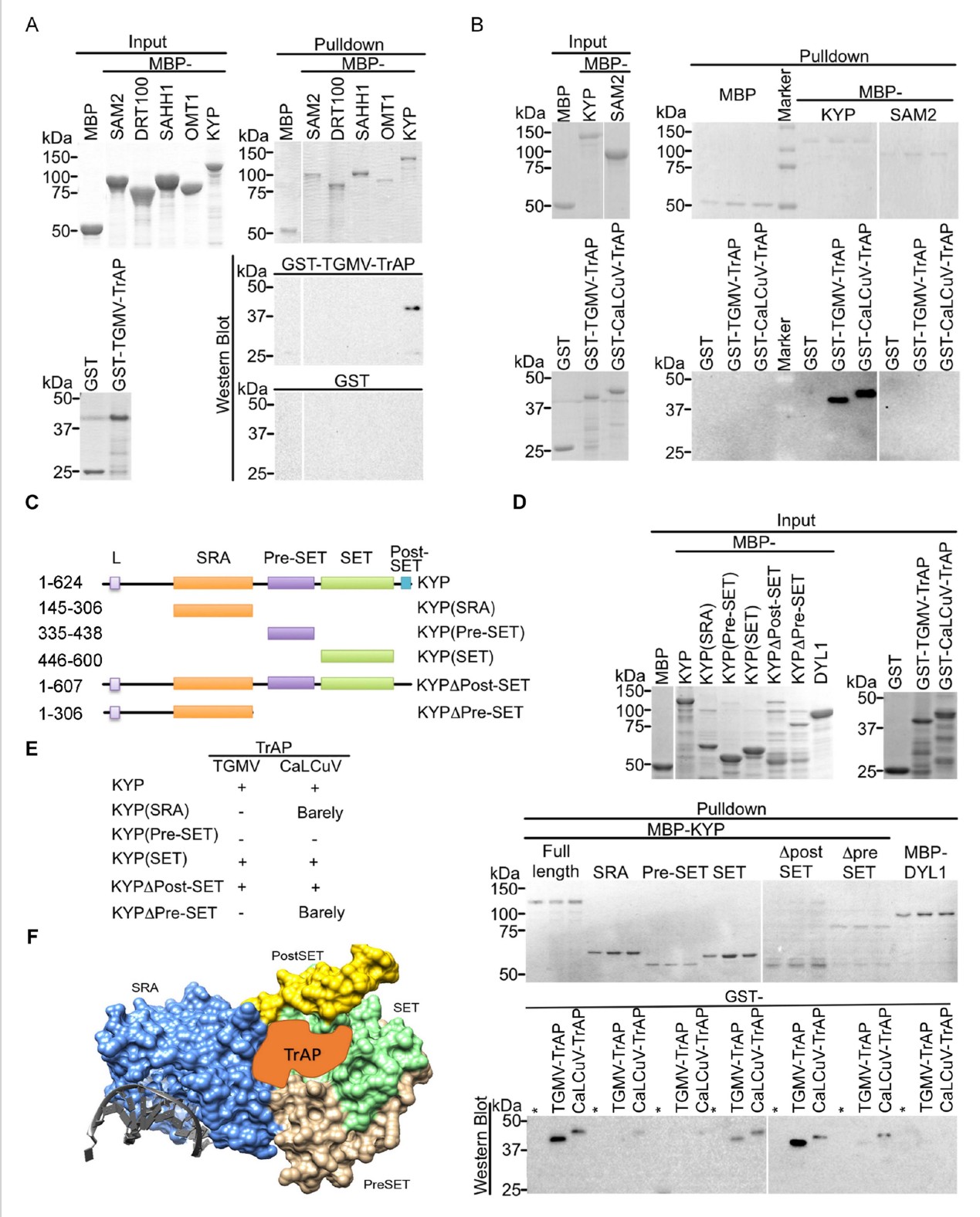

**Figure 4**. TrAP interacts directly with KYP through the SET domain. (**A**, **B**) In vitro pull-down assays showed that KYP specifically interacted with TGMV TrAP (**A**) and CaLCuV TrAP (**B**). Left panel, Coomassie brilliant blue R250 staining of the proteins shows their mobility. Right panel, output of in vitro pull-down assays. All His-MBP-tagged bait proteins and His-GST-tagged prey proteins were purified from *Escherichia coli* using Ni-NTA columns. In all assays, 2.5 μg of prey proteins were pulled down with the indicated bait proteins (2.5 μg each) using amylose resin. The recovered MBP-tagged bait proteins were

*Figure 4. Continued*

monitored by Coomassie brilliant blue R250 staining. All the experiments were done at the same time, and the samples were run in the same gels. The spacers in the images indicate digital rearrangements of the pictures. The output of the GST-tagged prey proteins was analyzed by western blot using a monoclonal anti-GST antibody. (**C**) Schematic diagram of full-length and truncated forms of KYP. The numbers on the left refer to the amino acid residues in KYP protein. Locations of SRA, pre-SET, SET, and post-SET domains are shown. (**D**) In vitro pull-down assays of truncated KYP proteins and TrAPs. The experiments were done as in (**A**, **B**). The GST negative control was loaded in the lanes marked with (*). (**E**) Summary of interaction between the truncated KYP proteins and TrAP encoded by TGMV and CaLCuV. (**F**) Model of possible KYP-TrAP interaction based on the experimental results from panels **D** and **E**. KYP structure was generated in Chimera from PDB: 4QEN data set (*Du et al., 2014b*), domains are color coded and indicated.

The following figure supplements are available for figure 4:

**Figure supplement 1**. TrAP does not interact with LHP1 in vitro.

**Figure supplement 2**. TrAP directly interacts with KYP paralogs SUVH2, 5, and 6.

(*Figure 3D*). Recent structural analysis on KYP revealed that SET and pre-SET domains constitute two modules: one forms a narrow pocket harboring the H3 tail (1–15aa), whereas the other binds the SAM cofactor, together with the post-SET domain (*Du et al., 2014a*). Because the post-SET domain does not seem to contribute to the KYP-TrAP interaction, TrAP could potentially occupy the histone-binding cavity (*Figure 4F*).

Since the SET domain is well conserved among histone methyltransferases (HMTase) (*Liu et al., 2010*), we next investigated whether TrAP interacts with KYP paralogs. In vitro pull-down assays showed that TrAP indeed interacted with numerous tested HMTases (SUVH2, 5, and 6) (*Figure 4—figure supplement 2*). Notably, loss-of-function mutants of *SUVH2* display early-flowering phenotype, suggesting that TrAP might target this protein in vivo as well.

## TrAP inhibits the catalytic activity of KYP in vitro

The specific interaction of TrAP with the catalytic SET domain of the SUVHs prompted the question of whether TrAP affects KYP activity. To test this, we set up an in vitro reconstitution of H3K9 methylation using His-GST-KYP purified from *Escherichia coli* as the enzyme source, recombinant histone 3 as the substrate, and methyl-$^{14}$C-SAM as the methyl donor (*Figure 5A*). Under our experimental conditions, 1 µM GST-KYP methylated 3 µM of histone 3 in less than 5 min at 37°C, as detected by saturation of the radioactive signal (*Figure 5B*). Excitingly, incubation of His-GST-KYP with His-MBP-TrAP from either TGMV or CaLCuV reduced the initial velocity of KYP transmethylation activity in a dose-dependent fashion, whereas His-MBP alone did not affect KYP catalysis (*Figure 5B*). Quantification of signal intensity revealed that the TrAP-KYP molar ratio of 2 was enough to cause approximately 50% inhibition of KYP activity (half maximal inhibitory concentration [IC$_{50}$]) (*Figure 5C*). Thus, our results indicated that TrAP potently inhibited the catalytic function of the HMTase.

## TrAP reduces H3K9me2-repressive marks in vivo

Given that in vivo TrAP genetically interferes with the TGS pathway, and in vitro it physically interacts with HMTases (KYP, SUVH2, 5, 6) and inhibits the activity of KYP, we wondered whether TrAP alters KYP function in vivo. To address this question, we conducted chromatin immunoprecipitation (ChIP) analyses of H3K4me3, H3K9me2, and K3K27me3 marks (*Figure 6*; *Figure 6—figure supplement 1*; *Figure 6—figure supplement 2*) on numerous KYP-regulated TEs (*Figure 6A*) in TrAP overexpression plants as compared to Col-0, *ddm1*, *kyp*, and *lhp1* control plants. Remarkably, all tested loci in *TrAP* transgenic lines displayed consistent reduction of H3K9me2 and H3K27me3, whereas changes of H3K4me3 were variable. This molecular phenotype mimicked *kyp* mutant at all tested loci, but not *lhp1* mutant. This result indicated inhibitory effect of TrAP on KYP activity on the TEs in vivo.

Since TrAP is a transcriptional activator protein, we next asked whether the increased transcription of the *TrAP*-responsive protein-coding genes is associated with changes in the histone methylation status. First, we screened numerous loci that showed transcriptional deregulation in the *TrAP* overexpression lines (*Figure 2B–D,G,H*) for the presence of various histone modifications. We identified a dozen loci in which H3K9me2 marks are easily detected in wild-type plants. We then

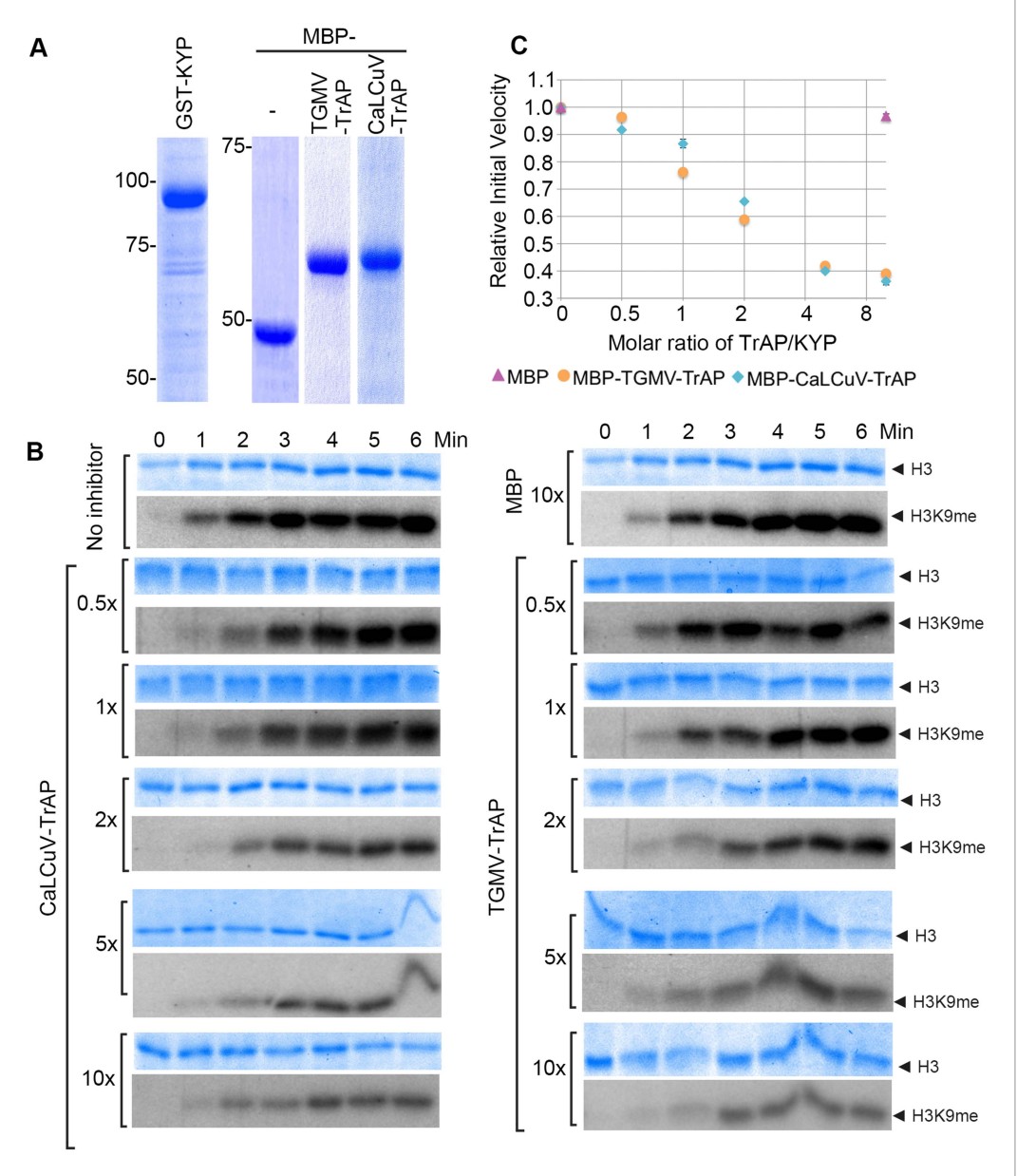

**Figure 5**. TrAP inhibited HMTase activity of KYP in vitro. (**A**) Coomassie staining of purified proteins uses for the assays. (**B**) In vitro HMTase reconstitution assays with different molar ratio of MBP and MBP-TrAP proteins (0–10✕) relative to GST-KYP. The recombinant KYP was incubated without (buffer only) or with the indicated proteins before addition of histone 3 and $C^{14}$-SAM. The reactions were done in a 6-min time course; aliquots were resolved in 18% SDS PAGE and stained with Coomassie blue R250 to show histone 3 input (top panels). The dried gels were auto-radiographed to detect $^{14}C$-methylated histone 3. (**C**). Plotting of KYP initial velocity vs TrAP/KYP molar ratio. The initial velocity was calculated from the slope of the linear range in the signal vs time plot for each assay, and then the values were normalized using the non-inhibitor control as a standard of 1 to obtain the relative initial velocity with standard deviation (SD) from at least three biological repeats. The relative initial velocity is plotted as a function of inhibitor:enzyme molar ratio in a logarithmic scale of base 2.

conducted ChIP-qPCR for these loci in the *TrAP* transgenic plants. As expected, H3K9me2 and H3K27me3 were reduced in most of the tested loci in *ddm1*, *kyp* and *lhp1* mutants, whereas H3K4me3 was enriched. These results are consistent with the generally antagonistic roles of H3K4me3 and H3K9me2 modifications. Specifically, six out of seven TrAP deregulated loci including the

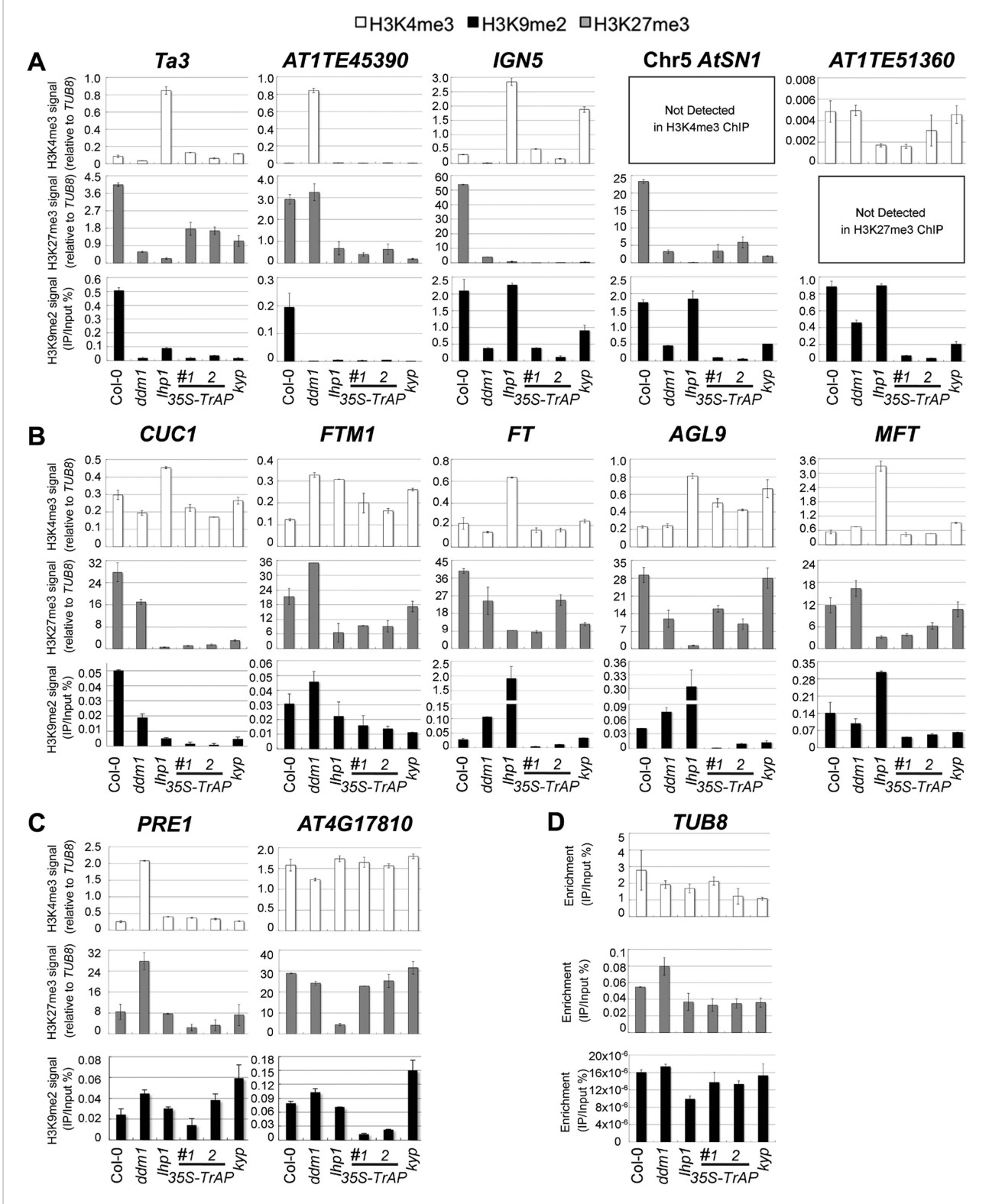

**Figure 6**. ChIP-qPCR analyses of H3K4me3, H3K9me2, and H3K27me3 in TrAP-regulated loci in vivo. (**A**) TrAP-activated transposons in heterochromatic regions contained reduced H3K9me2 and H3K27me3 but did not show consistent variation in H3K4me3 marks. (**B**) *TrAP*-upregulated flowering genes showed consistently reduced H3K9me2 and H3K27me3 marks compared to wild-type Col-0. (**C**) TrAP-downregulated genes displayed variable changes of H3K9me2 and H3K27me3 marks and no obvious changes of H3K4me3 mark. (**D**) Tubulin (*TUB8*) was used as internal control for all the ChIP experiments;
*Figure 6. continued on next page*

*Figure 6. Continued*

the percentage enrichment vs input is shown. ChIP assays were conducted on 11-day-old seedlings using antibodies specific for H3K9me2 (Abcam, Cat# ab1220), H3K27me3 (Millipore, Cat# 07-449), and H3K4me3 (Millipore, Cat# 04-745). Enrichment of H3K4me3 and H3K27me3 in each locus is normalized to that of *TUB8*; H3K9me2 enrichment is plotted as percentage of input. The standard deviation (SD) was calculated from at least three biological repeats.

The following figure supplements are available for figure 6:

**Figure supplement 1**. Western blot analysis to show specificity of antibodies used for ChIP assays in the study.

**Figure supplement 2**. ChIP-PCR assays for selected flowering genes and heterochromatic loci confirm ChIP-qPCR.

flowering-promoting genes displayed greater than twofold reduction in H3K9me2 and H3K27me3 in the *TrAP* overexpression plants compared to wild-type Col-0, while changes in H3K4me3 were inconsistent. This scenario was similar to that observed for TEs (*Figure 6B,C*). Collectively, the ChIP assays indicated that TrAP interferes with the epigenetic pathways through reducing repressive H3K9me2 marks in vivo.

## TrAP decreases CHH DNA methylation

KYP, SUVH5, and SUVH6 are required for maintenance of non-CG (CHG and CHH) methylation in *Arabidopsis* (*Stroud et al., 2013*; *Stroud et al., 2014*). We predicted that inhibition of KYP function by TrAP might indirectly cause reduction in non-CG DNA methylation. To test this, we conducted genome-wide bisulfite sequencing with 11-day-old seedlings of TrAP transgenic plants, *kyp* mutant, and Col-0. Consistent with previous studies, *kyp* mutant showed genome wide loss of methylation in CHG (~42.1%) and CHH (~21.7%), but not in CG (~2%) contexts when compared to Col-0. To our surprise, *TrAP* transgenic plants only exhibited decrease in methylation of CHH (~11%) but not CHG (*Figure 7A*).

To further analyze the effect of TrAP on DNA methylation, we identified differentially methylated regions (DMRs) by scanning the genomes in 200 bp tiles and comparing the level of methylation among *kyp* and *TrAP* overexpression plants with Col-0 ('Materials and methods') (*Supplementary files 5, 6, 7*). Given that non-CG methylation is highly co-localized and predominant in TE-rich heterochromatic regions (*Stroud et al., 2013*, *2014*; *Shen et al., 2014*; *Dubin et al., 2015*; *Yang et al., 2015*), we separated the DMRs into gene- and TE-rich regions (GRR and TERR respectively); When compared to Col-0, *TrAP* transgenic plants displayed loss of CHH methylation at GRR but not in TERR, nor in CG or CHG contexts (*Figure 7B*). Remarkably, of the 3442 and 1784 GRR hypomethylated DMRs identified in *kyp* mutant and TrAP transgenic plants, 1642 were shared. To better understand the effect of *TrAP* expression on CHH DNA methylation, the GRR DRMs were further separated into promoter, terminator, UTR, intronic, and coding regions. We found that *TrAP* hypomethylated DMRs in coding sequences, UTRs, and introns overlapped almost completely with the *kyp* mutant (96.7%, 96%, and 95.4% respectively) (*Figure 7C*). Notably, gene ontology analysis of the overlapped genes pointed to response genes, specifically in the protein kinase category (*Figure 7—figure supplement 1*; *Supplementary file 7*). This substantial overlap explains about one-third of the total CHH DMRs in *kyp* mutant, and one half of *kyp* DMRs in genic regions (*Figure 7C*). Together, these results suggest an inhibitory effect of TrAP in KYP-dependent CHH DNA methylation.

## TrAP inhibits KYP activity to counter host defense

In animals, H3K9 methylation promotes TGS and latency of integrated viruses. In plants, Geminivirus constitutes into a minichromosome that also undergoes epigenetic regulation. The specific TrAP-KYP interaction, inhibition of KYP activity in vitro, and reduction of H3K9me2 and CHH methylation in vivo suggest that KYP is a major factor in combating viruses. Previous studies showed that *kyp* mutants are hypersusceptible to Geminivirus infection (*Hanley-Bowdoin et al., 2013*); these experiments were reproducible in our hands (*Figure 8A–D*). Moreover, the hypersusceptibility of *kyp* mutant to the virus could be rescued by the wild-type *KYP* transgene under the control of both the native and the constitutive 35S promoters (*Figure 8A–D*); further validating a role for KYP in regulation of viral infection. In line with the phenotypic complementation, wild-type *KYP* almost completely rescued

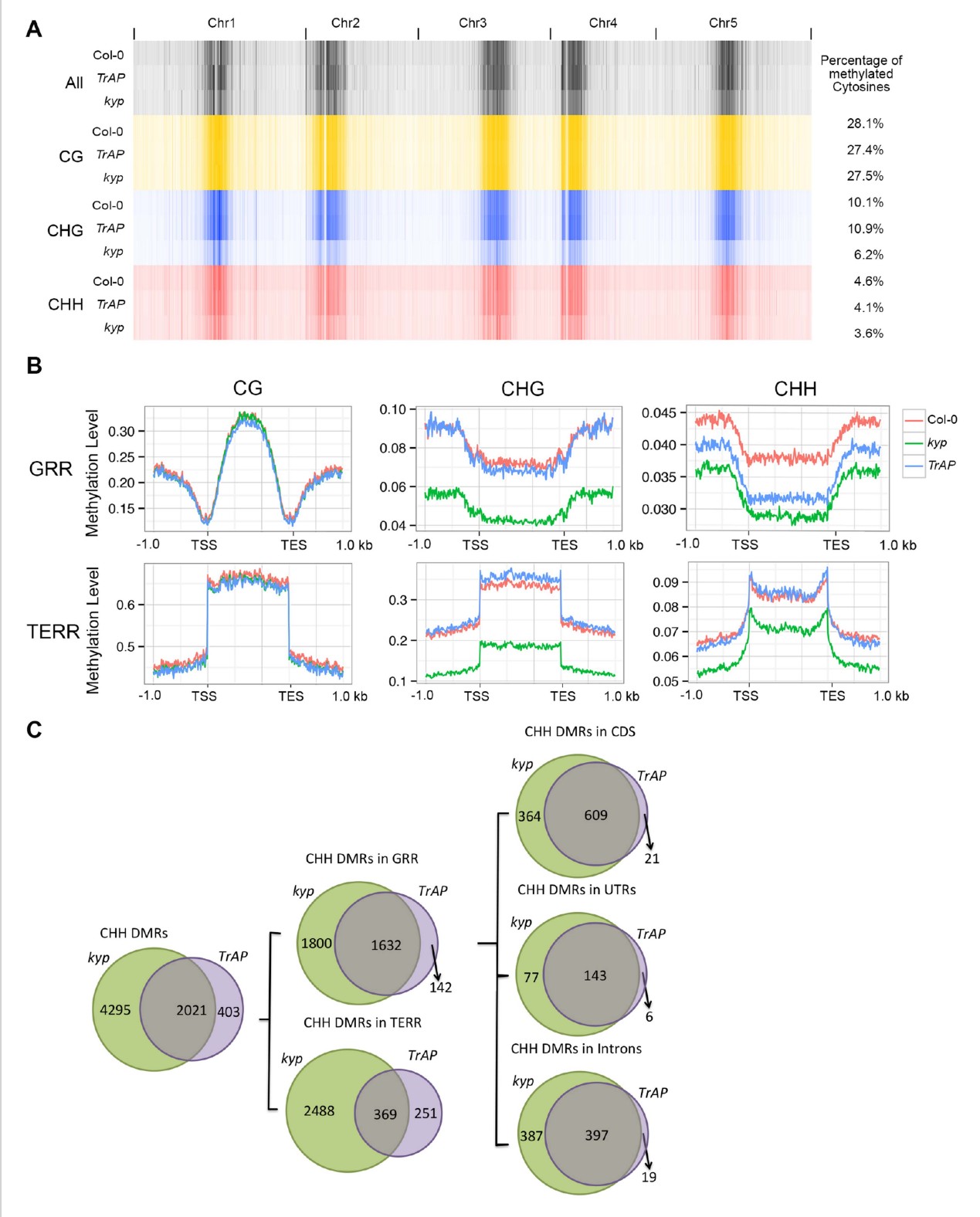

**Figure 7**. TrAP reduces CHH DNA methylation in vivo. (**A**) Genome-wide heatmap of DNA methylation levels in Col-0, *kyp* mutant, and *TrAP* transgenic plants (Left). Sequence context of all cytosine, CG, CHG, and CHH methylation was depicted as black, yellow, blue, and red, respectively. The percentage of cytosine methylation is shown for each genotype (Right). (**B**) Average distribution of context-specific DNA methylation in Gene- and TE-rich regions in *Figure 7. continued on next page*

Figure 7. Continued

Col-0 (red), *kyp* mutants (green), and *TrAP* transgenic plants (blue). (**C**). Overlap of CHH DMRs between *TrAP* transgenic plants and *kyp* mutant.

The following figure supplement is available for figure 7:

**Figure supplement 1**. Gene ontology of CHH hypomethylated genes in TrAP transgenic plants and kyp mutant.

H3K9me2 defects of the TEs in the *kyp* mutant (*Figure 8E*, lanes 1–4). Interestingly, although the accumulation of *KYP* transcripts was substantially increased when transcribed from the *35S* promoter when compared to the native promoter, the steady-state protein level was only twofold to threefold higher in the *35S-Flag-4Myc-KYP* than in the $P_{KYP}$-*Flag-4Myc-KYP* transgenic plants (data not shown). These observations suggest a possible homoeostatic regulation of this critical TGS component.

Geminivirus minichromosome harbors H3K9me2 (*Figure 8G*; *Figure 8—figure supplement 1*) (*Hanley-Bowdoin et al., 2013*). Interestingly, H3K9me2 marks were dramatically decreased on several, but not all tested loci in the viral genome in *kyp* mutant, and the methylation marks were further restored by the *KYP* transgene, suggesting that viral chromatin H3K9me2 is catalyzed by KYP. Notably, the H3K9me2 amount was substantially enhanced in the locus defined by Primer #9 in the transgenic *KYP* overexpression plants (*Figure 8G*). It is noteworthy that this locus harbors the promoter element for *TrAP* itself (*Shung and Sunter, 2009*). This result suggests that the tight control of *TrAP* expression likely determines the balance between the host and virus interactions. If so, the result might also explain the relatively milder viral symptoms of *35S-Flag-Myc4-KYP* lines compared to WT control or $P_{KYP}$-*Flag-Myc4-KYP* complementation lines (*Figure 8A–D*). To further test whether KYP methylated the viral chromatin, we performed ChIP assays using monoclonal anti-Flag antibody to pull down *Flag-Myc₄-KYP* bound chromatin (*Figure 8H*). Excitingly, KYP was found at all tested loci in the viral chromatin (*Figure 8I*). Notably, regions delineated by the primer #7 showed KYP-independent H3K9me2 marks but were still immunoprecipitated in the KYP-chromatin complex. A possible explanation is incomplete chromatin shearing of the small viral mini-chromosome under the conditions used in this experiment, which were standardized for host chromatin ChIP. Alternatively, this might suggest the presence of additional epigenetic regulation that masks KYP activity on this locus. This notwithstanding, our results indicate that KYP directly deposits the H3K9me2 mark on the Geminivirus minichromosome to reinforce the silent status of the virus.

Since constitutive expression of *TrAP* reduced H3K9me2 in vivo, we wondered if Geminivirus infection could also decrease the repressive marks in the host. To this end, we tested the KYP-controlled endogenous transposons for H3K9me2 accumulation. CaLCuV-infected plants showed 60–100% H3K9me2 loss in the tested loci compared to the amount in the mock-inoculated plants (*Figure 8E*), this is reminiscent of the molecular phenotype of *TrAP* transgenic plants, which showed lower enrichment of H3K9me2 mark at the studied loci (*Figure 6C*). These results indicated that Geminivirus infection largely removed the repressive H3K9me2 marks in the transposons, and that the removal resulted at least in part from TrAP function. Given that TrAP is known to activate the expression of viral genes in the minichromosome and that endogenous transposons can serve as a proxy for viral genomes, it is conceivable that virus-encoded TrAP acts to suppress KYP in order to prevent the deposition of H3K9me2-repressive marks in the epigenome, to activate the expression of viral genes.

## *kyp* mutants sustain systemic infection of CaLCuV lacking a functional *TrAP* gene

Previous studies show that TrAP is required for the accumulation of the virus infective form, single-stranded (ss) DNA (*Hayes and Buck, 1989*). TrAP is indispensable for systemic infection of Begomoviruses because it activates the expression of the viral ssDNA binding proteins (nuclear shuttle protein and coat protein), which are essential for releasing the virus from the nucleus and for cell-to-cell spreading (*Sunter and Bisaro, 1992*). If TrAP-mediated transcriptional activation and accumulation of ssDNA result from inhibition of KYP and correspondingly heterochromatin formation, then infectivity of CaLCuV lacking TrAP should be impaired in wild-type but recovered in *kyp* plants. To test this hypothesis, we engineered a CaLCuV variant without functional TrAP protein (CaLCuV Δ*trap*) by changing a single nucleotide (T to A) that produces an amber mutation and introduces an

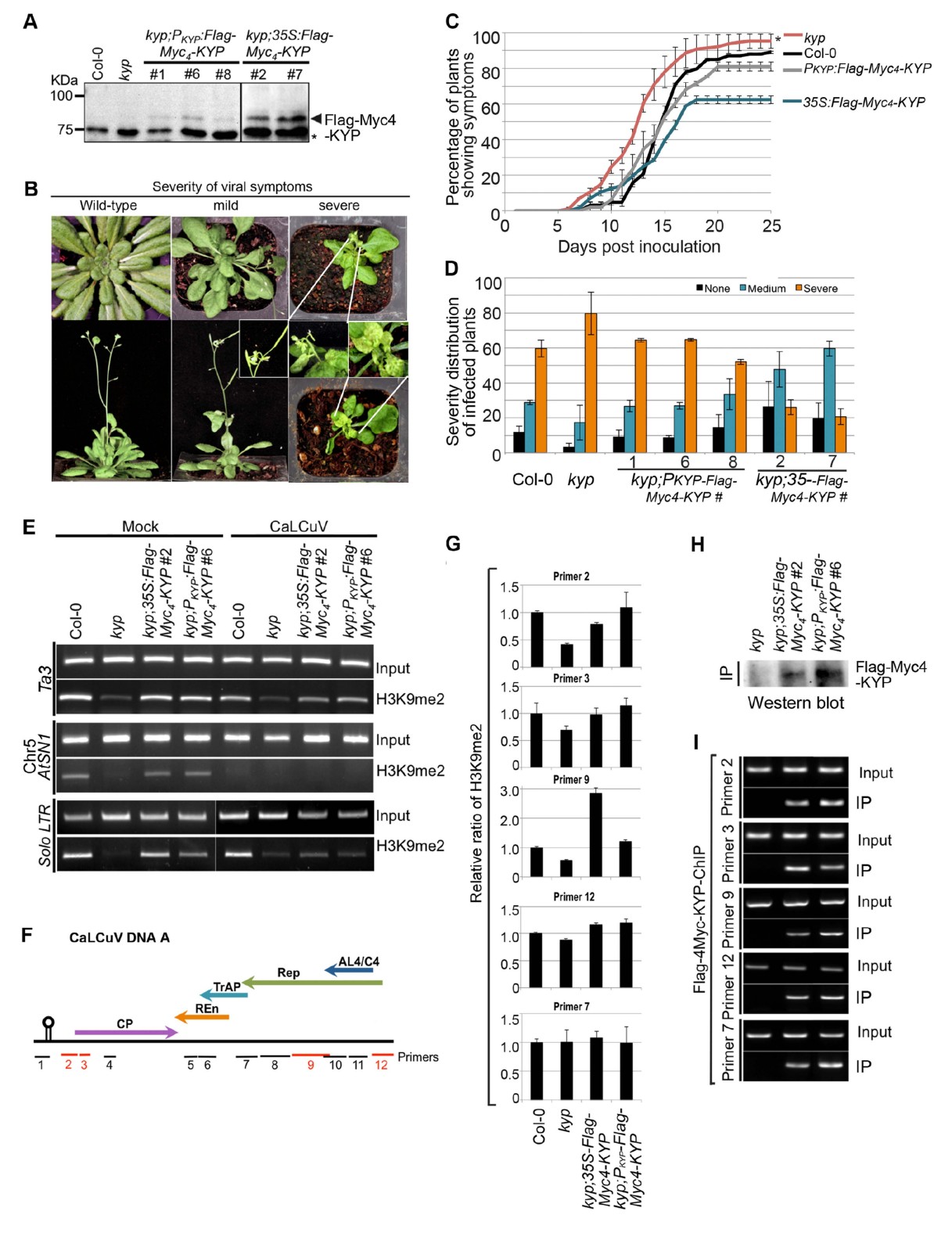

**Figure 8**. KYP methylates Geminivirus chromatin as a host defense. (**A**) Western blot analysis of *kyp* complementation lines expressing $P_{KYP}$- or *35S-Flag-4Myc-KYP* using anti-myc antibody. *, a cross-reaction band serves as a loading control. (**B**) Representative CaLCuV symptoms with different severities. (**C**) Time course of CaLCuV symptom development in *kyp* mutant and the complementation lines. The mean values were calculated with SD from at least three experiments (>30 plants/line). (**D**) CaLCuV symptom severity in Col-0, *kyp* mutant, and the complementation lines. The mean values were calculated

*Figure 8. continued on next page*

*Figure 8. Continued*

with SD from at least three experiments (>30 plants/line). (**E**) ChIP-PCR of H3K9me2 marks in heterochromatic loci in *kyp* mutant and complementation lines inoculated with mock or CaLCuV. Note: CaLCuV infection largely removed H3K9me2 marks from heterochromatic loci. (**F**) Schematic linearized representation of the regions of viral genome A selected for ChIP assays. (**G**) ChIP-qPCR assays showed KYP-dependent enrichment of H3K9me2 in several tested loci in the viral chromatin. The relative value of histone methylation in each sample was normalized to that of wild-type control where the signal was arbitrarily assigned a value of 1 with standard deviation (SD) from at least three biological repeats. Note: the region defined by # Primer 7 serves as a negative control. (**H**) Western blot analysis to detect Flag-Myc$_4$-KYP in the ChIP (IP) samples using anti-myc antibody. (**I**) ChIP-PCR assays showed that KYP binds to the viral minichromosome. The ChIP assays were done with a monoclonal anti-Flag antibody.

The following figure supplement is available for figure 8:

**Figure supplement 1**. Virus chromatin contains H3K9me2 marks.

XbaI restriction site after the sixth codon in the *TrAP* gene (*Figure 9A*). Then, we assessed systemic infection of CaLCuV and CaLCuV Δ*trap* on wild-type and *kyp* plants (*Figure 9B*). Consistent with the hypersusceptible phenotype of *kyp* mutants to CaLCuV (*Figure 8C,D*), these plants accumulated significantly higher titers of CaLCuV relative to wild-type control as both the replicative intermediate, open circle (OC), and the infective particle, SS DNA (*Figure 9C*). Consistent with previous reports, the wild-type plants did not show any symptoms of infection (*Figure 9E*) or systemic accumulation of CaLCuV Δ*trap*, as evidenced by Southern blot analyses and PCR (*Figure 9C*). Particularly, PCR amplification followed by XbaI digestion of the viral *TrAP* region in wild-type plants infected with CaLCuV Δ*trap* showed loss of the XbaI restriction site (*Figure 9—figure supplement 1*). Excitingly, the viral DNA was detected, in low amount but reproducibly, in *kyp* mutants. Further PCR amplification of the *TrAP* gene followed by XbaI digestion, confirmed that the detected viral DNAs in *kyp* mutants indeed came from CaLCuV Δ*trap* and not from a reversion of the mutation in the virus genome. These results show that *kyp* plants sustained systemic infection of CaLCuV Δ*trap*. Noticeably, the titers of accumulated CaLCuV Δ*trap* were much lower than the ones of CaLCuV, suggesting either redundant activity of *KYP* paralogs in host defense or necessity of additional functions of TrAP besides inhibition of KYP activity for the virus to achieve efficient infection.

## Discussion

TrAP was among the first viral suppressors identified to interfere with the TGS pathway. The prevailing model is that TrAP lowers the reservoir of transferable methyl groups by targeting ADK, a key enzyme in the SAM pathway. Here, we propose a novel model in which TrAP regulates TGS by directly targeting KYP (*Figure 9F*). Several pieces of evidence supported our notion: (1) TrAP genetically interfered with the TGS pathway (*Figure 2*); (2) TrAP directly interacted with KYP in vivo and also with other HMTases in vitro (*Figure 3*; *Figure 4*); (3) TrAP inhibited the catalytic activity of KYP in vitro (*Figure 5*); (4) TrAP reduced the repressive H3K9me2 marks in vivo (*Figure 6*), and correspondingly, reactivated numerous loci that are otherwise repressed by KYP (*Figure 2*); (5) TrAP decreased CHH methylation in gene-rich regions that are also regulated by KYP (*Figure 7*); (6) methylation of viral chromatin entailed KYP (*Figure 8*); (7) KYP bound the viral chromatin (*Figure 8*); and (8) *kyp* mutants but not wild-type plants sustain low systemic infection of CaLCuV lacking TrAP protein (*Figure 9*). To our best understanding, this is the first evidence that a viral protein directly suppresses HMTases in the host TGS machinery.

Given that TrAP protein could concurrently limit the upstream supply of the methyl groups and directly inhibit downstream enzymatic activity of KYP, Geminiviruses appear to have evolved sophisticated strategies to cripple the host TGS pathway. What would be the biological advantages of blocking the TGS pathways? In eukaryotes, chromatin appears to be a critical battleground for virus–host interaction. Animals use histone modifications to reinforce the latency of integrated viruses (*du Chene et al., 2007*; *Narasipura et al., 2014*; *Park et al., 2014*). Plant DNA viruses including Geminiviruses and pararetroviruses replicate as nuclear minichromosomes or episomes. Clearly, TGS functions as an immune system to control virus replication and the expression of viral genes, in a similar fashion as the repression of endogenous TEs and transposon remnants. In previous studies and here in our experiments, *kyp* mutants displayed hypersusceptibility to Geminivirus infection, which

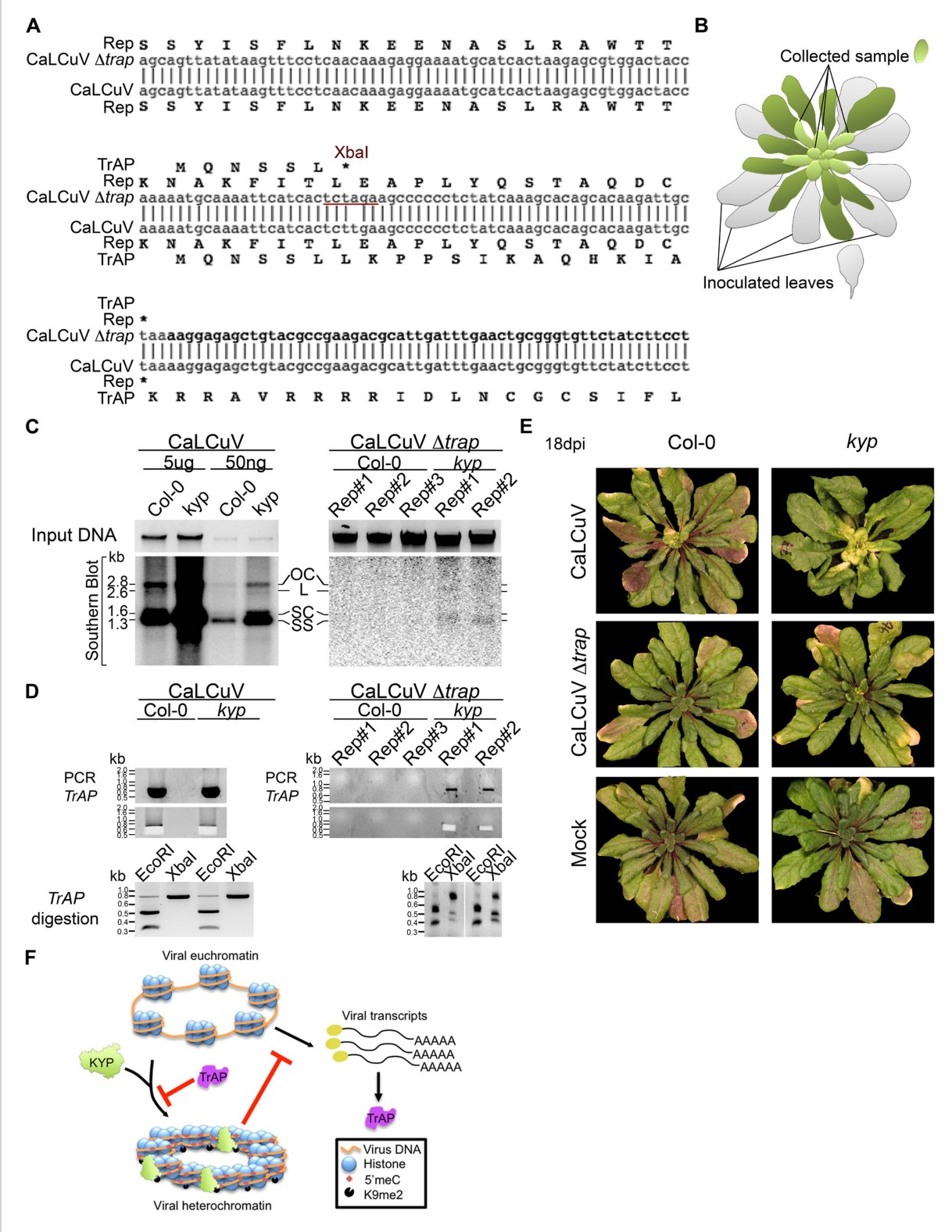

**Figure 9**. Infectivity of CaLCuV lacking functional TrAP protein. (**A**) Sequence alignment of CaLCuV Δ*trap* and CaLCuV sequences. The translated amino acids are shown for each sequence, and the XbaI restriction site resulting from the T to A point mutation is highlighted. (**B**) Schematics of the systemic infection experiment. Plants with eight true leaves (depicted in gray) were inoculated with the begomovirus, and 18 days post inoculation nine to 11 newly emerged, not inoculated, rosetta leaves (depicted in bright green) were collected to test for virus systemic infection. (**C**) Southern blot analysis of viruses in

*Figure 9. continued on next page*

*Figure 9. Continued*

non-inoculated leaves of infected plants. Ethidium bromide staining of total genomic DNA serves as a loading control (top panels). Southern blots were probed against CaLCuV DNA A common region (CR) (bottom panels); the viral populations are indicated as the replicative intermediate open circle (OC), linear (L), super coiled (SC) and the infective particle ss DNA. (D) Genotypic confirmation of the systemically amplified viruses. Top panels show PCR amplification of a *TrAP*-containing region; bottom panels show EcoRI and XbaI digestions of PCR products to examine the presence of the amber mutation in the gene. (E) Exemplary phenotypes of wild-type and *kyp* mutant plants inoculated with mock, CaLCuV and CaLCuV Δ*trap*. (F) Model of TrAP suppression of KYP activity to prevent epigenetic silencing of the viral chromatin. Geminivirus genome is packed on histone octamers to form a minichromosome. The minichromosome undergoes extensive H3K9me2 modification deposited by host-encoded KYP, and this modification could be further reinforced by DNA methylation, leading to formation of viral heterochromatin. As a counter-defense strategy, Geminivirus-encoded TrAP protein inhibits KYP activity to maintain the euchromatic status of the minichromosome to allow active replication and transcription of viral genes and correspondingly to escape host surveillance.

The following figure supplement is available for figure 9:

**Figure supplement 1**. CaLCuV lacking functional TrAP protein cannot cause systemic infection in wild-type plants.

could be rescued by exogenous wild-type *KYP* gene. KYP directly acts on viral chromatin and deposits the repressive H3K9me2 mark on the viral chromatin. Moreover, viruses lacking the essential TrAP protein can, although inefficiently, cause systemic infection in *KYP*-deficient hosts. All these facts point out the unambiguous role of KYP-mediated TGS in defense against viral infection. On the other hand, TrAP functions to inhibit KYP catalytic activity, reducing the repressive H3K9me2 mark, to activate transcription of viral genes. Our KYP reconstitution assays show that TrAP:KYP molar ratio of around 2 is enough to cause ~50% inhibition of KYP activity, indicating that TrAP is a potent inhibitor of HMTase activity. Thus, direct inhibition of KYP represents a novel counter-defense mechanism for virus survival in the hosts. This mechanism could account for the long-documented essential role of TrAP in expression of viral genes including the coat protein and the nuclear shuttle protein (*Sunter and Bisaro, 1992*; *Yang et al., 2007*; *Shen et al., 2009*; *Hanley-Bowdoin et al., 2013*).

In plants, H3K9me2 marks often correlate with non-CG DNA methylation, and in particular with CHG modification. The tight coordination results from a self-enforcing loop consisting of KYP and CMT3 (*Du et al., 2014a*). Briefly, KYP methylates H3K9 to generate the binding sites for CMT3, which further methylates CHG DNA to create more binding sites for KYP. Consequently, the crosstalk between DNA and histone methylation ascertains the silent status of heterochromatin. We hypothesized that TrAP targets KYP to reduce H3K9me2 marks, and this inhibition would further decrease DNA methylation both on the host and on the viral genome. Here, we found that ectopic expression of TrAP had no effect on CHG but only on CHH methylation, at sites that are up to 96.7% overlapped with the hypo-methylated regions in *kyp* mutant (*Figure 7C*). In plants, sequence contexts of CHG and CHH methylation are largely overlapped throughout the genome and are maintained to a limited extent by all non-CG methyltransferases (*Stroud et al., 2014*). One outstanding question would be how could TrAP differentially alter methylation of CHH rather than CHG, given that TrAP targets KYP? Methylation in the CHH context is catalyzed by CMT2 and DRM1/2 in *Arabidopsis*, and the sites regulated through the two sets of enzymes are mostly non-overlapping. Thus, the functions of CMT2 and DRM1/2 are mostly non-redundant at CHH sites (*Stroud et al., 2013, 2014*). DRM1/2 largely catalyze CHH methylation at the TE-rich regions (*Stroud et al., 2014*). In our study, we did not observe any effect of TrAP on CHH methylation in the TEs, suggesting that TrAP might not (or not sufficiently) interrupt the RdDM pathway that entails DRM1/2 and 24-nt siRNAs (*Stroud et al., 2014*). Importantly, it has been recently discovered that bulk CHH methylation is maintained by CMT2 (*Shen et al., 2014*; *Stroud et al., 2014*; *Dubin et al., 2015*) and that its activity is largely dependent on H3K9me2. In this scenario, CMT2 recognizes methylated H3K9 but preferentially binds to di-methylated over mono-methylated histone tails. This preference towards H3K9me2 is not observed in the CHG methyltransferase CMT3, which can equally bind to all forms of H3K9 methylation (*Stroud et al., 2014*). We envision that methylation at CHH sites could be more sensitive to changes in H3K9me2 than CHG, since CMT3 could still maintain CHG DNA methylation in the presence of H3K9me1 resulting from residual activity of HMTase, as would be the case when KYP is inhibited by TrAP. Alternatively, CHH methylation might play an important but yet unappreciated regulatory role

in host defense genes (*Figure 7—figure supplement 1*). If so, preferentially targeting these loci by TrAP protein might represent a new counter-defense mechanism.

In the host, Geminivirus DNAs also undergo extensive methylation modification (*Raja et al., 2008*). In our study, TrAP reduces CHH methylation in the host and possibly in the viral genome, too. Remarkably, the Geminivirus replicase, Rep/AC1/C1 and the embedded protein AC4/C4, down-regulate the expression of host DNA methyltransferases *MET1* and *CMT3* (*Pumplin and Voinnet, 2013*; *Rodriguez-Negrete et al., 2013*); hence, it interrupts the reinforcing loop of histone and DNA methylation. Consistent to this study, *cmt3* mutants exhibit hypersusceptibility to viral infection (*Raja et al., 2008*). The fact that Rep/C4 repress *CMT3* expression in the host is in perfect alignment with our ChIP assay results on CaLCuV-infected plants, where the loss of H3K9me2 is even more severe than the observed in TrAP transgenic plants. Thus, interruption of the compelling feedback loop of histone and DNA methylation represents an important strategy to sustain transcription of viral chromatin. Together, it seems that the synergistic inhibition of histone and DNA methyltransferases by Geminivirus proteins evolves as a powerful tactic to win the arms race between host and pathogen.

Our in vitro assays clearly demonstrated that TrAP predominantly binds to the catalytic domain of KYP and inhibited its enzymatic activity. Whether TrAP might alter KYP conformation or block the accessibility of substrates to the active sites upon interaction awaits future structural analysis. We note, however, that *Arabidopsis* has 49 SET domain containing proteins of which 31 are considered to have HMTase activity (*Liu et al., 2010*). Arising from this fact is whether TrAP specifically targets KYP or promiscuously acts on additional HMTases. Although TrAP overexpression plants display molecular features in common with *kyp* mutant (i.e., reduced H3K9me2 levels), and an early flowering phenotype similar to loss-of-function mutant of the *kyp* paralog, *SUVH2*, the transgenic plants are phenotypically different from *kyp* single mutant. This could be due to functional redundancy between *KYP* and its paralogs in vivo; alas, the morphological phenotype of the higher-order mutants has not been fully documented.

We also noticed that *TrAP*-overexpression plants are morphologically similar to *lhp1*, which is characteristic of the H3K27me3 pathway. Moreover, the substantial overlapping of *TrAP*-responsive genes with *lhp1*-regulated genes strongly suggested that TrAP might target the *LHP1*-related H3K27me3 pathway (*Zheng and Chen, 2011*). Indeed, most of our tested loci in the host genome exhibited decreased H3K27me3 levels, consistent with the fact that ~50% genes were co-regulated by TrAP and *LHP1* in a genome-wide scale. Interestingly, recent ChIP–chip studies revealed that H3K27me3 and *LHP1*-bound sites are predominantly distributed in the euchromatic regions (*Turck et al., 2007*; *Zhang et al., 2007a*, *2007b*). This distribution is not correlated with KYP-dependent H3K9me2 marks that are highly enriched at pericentromeric regions as large and uninterrupted heterochromatic blocks (*Liu et al., 2010*; *Black et al., 2012*; *Du et al., 2014a*; *Liu et al., 2014*). H3K9me2 can also occur in euchromatic regions but rather exist as small heterochromatin patches (*Zheng and Chen, 2011*). In our current study, we did not further examine whether TrAP physically targets H3K27me3 HMTases in the *LHP1* pathway. But it is plausible that TrAP inhibits both KYP and *LHP1* pathways. Notably, we did not observe the repression of transcriptionally active histone methylation marks, such as H3K4me3 and H3K36me3. This observation suggests that TrAP does not target the HMTases that are writers for the active marks. How can TrAP distinguish KYP from those active writers remains unclear, but we propose that structural components other than the SET domain might contribute to the recognition and affinity of TrAP binding to HMTases (*Figure 4*). Alternatively, additional cellular factors might also contribute to the specificity.

The direct consequence of TrAP-dependent inhibition of KYP activity is to activate viral transcription and replication. Because KYP is a key effector of TGS in the host and regulates a broad array of endogenous genes, interference with this core component would reprogram the expression profile of the host genome and thus trigger a series of downstream cascade signaling events that impact the balance of host/virus interaction. As an example, TrAP suppresses auxin and cell growth, whether this change might constitute a defense mechanism for the host benefit or create a favorable cellular niche for virus propagation remains for further investigation.

In conclusion, our results support the notion that Geminivirus-encoded TrAP protein interferes with the TGS pathway and abrogates epigenetic silencing by direct interaction with KYP and inhibition of its transmethylation activity. Thus, Geminvirus TrAP functions clearly different from most of previously characterized viral suppressors, which target various steps of the PTGS pathway (*Ding and Voinnet, 2007*). Together with previous studies (*Raja et al., 2008*), we provide evidence that KYP evolves as a

critical immune system to control invading nucleic acids in plants; this is reminiscent of the roles of human SUV39H1 in maintaining the latency of HIV (*du Chene et al., 2007*), Epsteinn-Barr virus (Imai et al., 2014), and some other mammalian viruses. Thus, sequestering or interfering with this core component is an effective strategy for Geminivirus to block TGS and to subvert host defense; we expect this strategy to be used by other suppressors in plants and mammals. Given that many viral suppressors interrupt different steps in the PTGS pathways to efficiently combat host surveillance, it would not be surprising that additional key cellular factors in the TGS pathway, besides HMTase, were readily targeted by invading viruses in eukaryotes. Importantly, it is believed that compounds or drugs that alter chromatin methylation might ultimately be the most effective means of combating disease. Our discovery that TrAP inhibits a histone-modifying enzyme also offers a new natural strategy to develop epigenetic-targeted drugs to cure human diseases that arise from epigenetic dysfunction (*Højfeldt et al., 2013*) or to engineer new biotechnological products to improve agricultural productivity.

## Materials and methods

### DNA constructions

All the plant constructs were made using the Gateway system (Invitrogen, Carlsbad, CA, United States) (*Zhang et al., 2005*). The destination vectors (containing the destination cassette–DC-) pHyg-DC-CFP, pBA-DC-CFP, pBA-DC-3HA, pBA-DC, pBA-Flag-4Myc-DC, and pER10-YFP-DC (*Zhang et al., 2006b*) were used for transient expression in *N. benthamiana* and for stable *Arabidopsis thaliana* transformation. The vector pER10 for betaestradiol-inducible expression under the XVE promoter (*Zuo et al., 2000*) was modified to obtain the destination vector pER10cLUC-3HA-DC. For this, the pER10 vector was linearized with XhoI, filled in with Klenow fragment; and further digested with PacI. In parallel, the cLuc-3HA-DC insert was obtained by linearizing the pCambia1300-cLuc-3HA-DC (*Zhang et al., 2011*) plasmid treated with SacI/Klenow treatment and further digested with PacI. The vector and insert were ligated and transformed into *E. coli* DB3.1 and finally confirmed by sequencing. This vector was used together with pCambia Myc-DC-nLUC (*Zhang et al., 2011*) for transient expression in *N. benthamiana*.

The cDNA or DNA fragments were cloned into pENTR/D vectors, confirmed by sequencing, and then transferred to the appropriate destination vectors by recombination using the LR Clonase (Invitrogen). The primers for the cloning are listed in the *Supplementary file 8*. To drive KYP expression from its native promoter, we amplified a 2.7-Kb genomic region immediately upstream of the *KYP* start codon with the primers $P_{KYP}$ EcoRV for and $P_{KYP}$ BamHI rev. The binary vector pBA002a Flag-4Myc-KYP was obtained by the Gateway system using LR clonase. The resultant plasmid and PCR product harboring *KYP* promoter were digested with EcoRV/BamHI and ligated with T4 DNA ligase. The plasmid was confirmed by sequencing using the primer $P_{KYP}$ seq For.

For expression of recombinant proteins, the cDNA or DNA fragments were cloned into pMCSG9 or pMCSG10 vectors to produce His-MBP- or His-GST-tagged proteins respectively, by ligation independent cloning (*Eschenfeldt et al., 2009*) using primers that include 18 nt identical to the ends of the linearized pMCSG vector (*Supplementary file 8*). The pMCSG plasmids were linearized with the blunt end restriction enzyme SspI, and the sticky ends were generated by T4 DNA Polymerase (NEB, Ipswich, MA, United States) supplemented with 2.5 mM dGTP. In parallel, the complementary sticky ends in the PCR products were generated supplementing the T4 DNA Polymerase (NEB) with 2.5 mM dCTP. The mixture of treated plasmid:PCR (3:4, respectively) was incubated on ice for 30 min to allow annealing of the complementary free strands and transformed in *E. coli* DH5α. The plasmids were further confirmed by sequencing and transformed into *E. coli* BL21 Rossetta DE3 for expression.

To engineer the CaLCuV Δ*trap* DNA A infective clone the pNSB1090 plasmid was subjected to site directed mutagenesis by amplification with the primers CaLCuV_AL2_null_XbaI_for and CaLCuV_AL2_null_XbaI_rev (*Supplementary file 8*), digested overnight with DpnI, cleaned with the QIAquick PCR Purification Kit (Qiagen, Germantown, MD, United States), and amplified in *E. coli* DH5α. The plasmid was confirmed by sequencing and transformed in *Agrobacterium tumefasciens* ABI for the virus infection assays.

### Transgenic plants

*A. thaliana* (Col-0) plants were transformed with binary vectors by the floral-dip method (*Clough and Bent, 1998*; *Zhang et al., 2006a*). The transgenic seeds were selected on standard MS medium

(*Murashige and Skoog, 1962*) containing the appropriate selective agents: 10 mg/l glufosinate ammonium (Sigma-Aldrich, St. Louis, MO, United States) or 25 mg/l hygromycin (Sigma), together with 100 mg/l carbenicillin (Sigma). *kyp* mutant (SALK_130630C) was obtained from the Arabidopsis Biological Resource Center (ABRC) and confirmed by genotyping and qRT-PCR.

## Expression and purification of recombinant proteins

For in vitro pull-down or HMTase reconstitution assays, the recombinant proteins were purified following either one or two-step affinity purification procedure. His-MBP-tagged proteins were first purified through Immobilized Metal Affinity Chromatography (IMAC) using the Ni-NTA resin (Qiagen), followed by amylose resin (NEB) according to manufacturers' protocols. His-GST-tagged proteins were initially purified through sepharose glutathione column (GE Healthcare, United Kingdom) followed by IMAC. Specifically for HMTase assays, His-MBP-CaLCuV_TrAP, His-MBP-TGMV_TrAP, and His-MBP were prepared in lysis buffer (50 mM Tris-HCl pH 9, 300 mM NaCl, 10 mM 2-mercaptoethanol, 2 mM PMSF), incubated with the Ni-NTA resin at 4°C for 1 hr, eluted with 300 mM imidazole and immediately incubated with the amylose resin (NEB) at 4°C for 1 hr. The proteins were eluted with 10 mM maltose and the elute was further separated by size exclusion chromatography (SEC) in column buffer (20 mM Tris-HCl pH 9, 100 mM NaCl); the fractions containing the target protein were pulled together, concentrated to 100 μM, aliquoted and stored at −80°C until usage. His-GST-KYP was prepared in PBS buffer (140 mM NaCl, 2.7 mM KCl, 10 mM $Na_2HPO_4$, 1.8 mM $KH_2PO_4$, pH 7.3, 10 mM 2-mercaptoethanol) incubated with the sepharose glutathione 1 hr at 4°C, eluted with elution buffer (50 mM Tris-HCl pH 9, 200 mM NaCl, 10 mM reduced glutathione, 10 mM 2-mercaptoethanol). The elute was further purified through the Ni-NTA column and finally through SEC in column buffer (20 mM Tris-HCl pH 9, 100 mM NaCl); the fractions containing the target protein were pulled together, concentrated to 25 μM and aliquoted for usage.

For in vitro pull down assay, both the prey (His-GST-CaLCuV-TrAP, His-GST-TGMV-TrAP, and His-GST) and the bait (His-MBP-Bait) proteins were purified by IMAC using the lysis buffer (50 mM Tris-HCl pH 8, 300 mM NaCl, 20 mM imidazole, 2 mM PMSF), incubated with the Ni-NTA resin at 4°C for 1 hr, eluted with 300 mM imidazole and immediately dialyzed in storage buffer (20 mM Tris-HCl pH 8, 150 mM NaCl, 2 mM 2-mercaptoethanol, 50% Glycerol) at 4°C overnight.

## In vitro pull down and Co-IP assays

In vitro pull-down assays and in vivo Co-IP were done as described (*Zhang et al., 2005*). Briefly, 2.5 μg of 6His-GST-tagged prey proteins were pre-absorbed to 50 μl of the amylose resin (NEB) for 1 hr at 4°C in 1 ml of binding buffer (50 mM Tris-HCl pH 7.5, 150 mM NaCl, 0.2% glycerol, 0.6% Triton X-100, 0.5 mM 2-mercaptoethanol, 2 mM PMSF). The proteins were recovered by ultracentrifugation at 12,000×*g* for 2 min, transferred to a second tube containing 2.5 μg of the MBP-tagged bait protein, and incubated at room temperature for 2 hr. The protein complexes were harvested by adding 50 μl amylose resin beads, followed by 2 hr incubation at room temperature, and cleaned with six vigorous washes with buffer. The pulled-down proteins were resolved by SDS-PAGE and the preys were detected by western blot using anti-GST antibody.

For Co-IP experiments, *N. benthamiana* leaves were collected 2 days after agroinfiltration, ground in liquid nitrogen and stored at −80°C until use. For the assay, total proteins were extracted from 0.4 g of ground powder in 1.2 ml (3 vol) of IP buffer (40 mM Tris-HCl pH 7.5, 300 mM NaCl, 5 mM $MgCl_2$, 2 mM EDTA, 4 mM DTT, 0.5% Triton X-100, 1 mM PMSF, 5% glycerol, 1 pellet/25 ml Complete EDTA-free protease inhibitor [Roche, Indianapolis, IN, United States]); then, the soluble proteins were cleared twice by ultracentrifugation at 20,000 × rcf for 15 min at 4°C. The protein complexes were immunoprecipitated with 15 μl Anti-c-Myc-agarose affinity gel (Sigma–Aldrich #A7470) at 4°C for 2 hr, the unspecific-bound proteins were removed by four consecutive washes with the IP buffer with 10 min incubation each at 4°C. The protein complexes were eluted in 200 μl of elution buffer (5 mM EDTA, 200 mM $NH_4OH$) for 20 min. The supernatant was collected, frozen in liquid nitrogen and dried using the Savant SpeedVac concentrator; finally, the sample was solubilized in 50 μl of 2× SDS-loading buffer for western blot analyses.

## Two-step immunoprecipitation for mass spectrometry analysis

9-day-old wild-type control and *Arabidopsis* transgenic plants expressing *XVE-Flag-Myc4-TrAP* were induced for 16 hr with 25 μM ß-estradiol in liquid MS media, ground in liquid nitrogen and stored at

−80°C until use. For the assay, total proteins were extracted from 10 g of ground powder in 40 ml (4 vol) of IP buffer (20 mM Tris-HCl pH 7.5, 150 mM NaCl, 4 mM MgCl$_2$, 50 µM ZnCl$_2$, 0.1% Triton X-100, 1 mM PMSF, 1% glycerol, 1 pellet/17 ml Complete EDTA-free protease inhibitor [Roche], 15 µM MG132); then, the soluble proteins were cleared twice by ultracentrifugation at 20,000 × rcf for 15 min at 4°C. The protein complexes were first immunoprecipitated using 500 µl of Anti-FLAG M2 magnetic beads (Sigma-Aldrich, Cat# M8823) and incubated in slow rotation for 2 hr at 4°C, the nonspecific-bound proteins were removed by three consecutive washes with 15 ml of IP buffer for 10 min incubation each at 4°C. The protein complexes were then eluted by competition with 100 mg/ml FLAG peptide and subsequently immunoprecipitated with 100 µl Anti-c-Myc-agarose affinity gel (Sigma–Aldrich #A7470) at 4°C for 1.5 hr, the nonspecific-bound proteins were removed by five consecutive washes with the IP buffer with 5 min incubation each at 4°C. The protein complexes were eluted in 200 µl of elution buffer (5 mM EDTA, 200 mM NH$_4$OH) for 20 min. The supernatant was collected, frozen in liquid nitrogen and dried using the Savant SpeedVac concentrator; finally, the sample was solubilized in 30 µl of 2× SDS-loading buffer and run in 10% SDS-PAGE. The samples were run to one-third of the gel, stained with Coomassie blue and collected by excising the whole lane for mass spectrometry analysis in the Taplin Mass Spectrometry Facility at Harvard Medical School.

## Southern blot analyses

The plant material was lysed in CTAB buffer (100 mM Tris HCl pH 8.0, 20 mM EDTA pH 8.0, 1.4 M NaCl, 2% CTAB, 2% ß-mercaptoethanol); then total DNA was extracted with phenol:chlorophorm:isoamyl alcohol (25:24:1) and precipitated with 2-propanol. The DNA was treated with RNase A and further purified with phenol:chlorophorm:isoamyl alcohol (25:24:1) and precipitated with ethanol, then dissolved in ultrapure water. The specified amount of DNA was separated by electrophoresis in 0.8% agarose, transferred overnight by capillarity to a Hybond-N membrane (GE Healthcare), and probed with $^{32}$P-labeled probe targeting the CR region of CaLCuV DNA A. The probe was obtained by PCR using the primers CR_F and CR_R (*Supplementary file 8*) and labeled using the Rediprime II DNA Labeling System (GE Healthcare) following the manufacturer's instructions.

## RNA blot analyses

Total RNA was extracted using Trizol reagent from either adult rosette leaves or 2-week-old seedlings of independent transgenic lines, the RNA blots were then performed as previously described (*Zhang et al., 2006b*).

## Immunoprecipitation of Flag-AGO1-associated small RNAs

Immunoprecipitation of Flag-AGO1-associated small RNAs were performed as described (*Zhang et al., 2006b*). RNA was recovered with Trizol reagent from the immunoprecipitates, separated in 8 M urea, 15% polyacrylamide gels and subjected to RNA blot analysis of low-molecular-weight RNAs.

## Luciferase complementation imaging (LCI) assay

The LCI was performed on 4-week-old *N. benthamiana* leaves infiltrated with various combinations of *A. tumefaciens* GV3101 harboring pCambia Myc-TrAP-nLUC or pCambia Myc-nLUC and *A. tumefaciens* ABI carrying pER10cLUC-3HA or pER10cLUC-3HA-candidate proteins. The agrobacteria containing the pER10 plasmids were incubated with 25 µM beta-estradiol for 3 hr prior infiltration, and all the cultures were adjusted to OD$_{600}$ = 0.8. The transfected leaves were assayed 2 days after agroinfiltration by adding the substrate (10 mM luciferin). The sprayed leaves were incubated in total darkness for 5 min and photographed using an electronmultiplying charge-coupled device (EMCCD) camera, Cascade II 512, from Photomerics (Roper Scientific, Trenton, NJ, United States). The images were processed with WinView32 Ver 2.5.19.7 (Roper Scientific).

## Confocal microscopy and FRET assays

Leaves of 4-week-old tobacco plants (*N. benthamiana*) were agroinfiltrated with syringe without needle as previously described (*Zhang et al., 2005*) with *A. tumefasciens* ABI carrying pBA-TrAP-CFP and pER10-YFP-Test protein. The agrobacteria containing the pER10 plasmids were incubated with 25 µM beta-estradiol for 3 hr prior infiltration, and all the cultures were adjusted to OD$_{600}$ = 0.8. The plants were maintained for 2 days at 24°C (16 hr light/8 hr dark). The co-localization was evaluated

using a Nikon inverted microscope Eclipse T$i$-E (Nikon, Japan). CFP signal was measured by excitation with Shutter 10-3 filter 3 (CFPHQ [Ex]), and emission was detected at 485 nm; YFP used Shutter 10-3 filter 4 (YFPHQ [Ex]) and emission was detected at 540 nm. The images were processed using NIS-Elements-AR 4.30.01 (Nikon) and Adobe Photoshop software.

FRET-AB experiments were performed on $N.$ $benthamiana$ epidermal cells of 4-week-old leaves agroinfiltrated with a 1:1 mixture of pBA-TrAP-CFP and pER10-YFP-KYP to a final $OD_{600} = 0.8$. YFP and CFP signals were captured with a Zeiss LSM 710 confocal Microscope (Zeiss, Germany). FRET was determined by the acceptor photobleaching method (Kenworthy, 2001; Daelemans et al., 2004). First, to define the base line, the signal intensities of a pre-photobleach CFP (donor) and YFP (acceptor) were acquired by exciting with the 458 and 514-nm laser lines, respectively. Then, three regions of interest in the cell were selected: #1, Autofluorescence control; #2, non-photobleaching control; and #3, FRET-AB region. The CFP donor was excited with the 458 nm laser line for all FRET experiments; the emission of both CFP and YFP was recorded at 485 nm and 540 nm. Regions #1 and #3 were rendered free of YFP by consecutive cycles of bleaching recovery with the 514-nm laser line until no recovery of YFP was detected. The CFP and YFP signals were monitored throughout the experiment. After correction for background with control region #1 and for photobleaching of the donor because of imaging with control region #2, the FRET efficiencies (E) in the region #3 was calculated from the CFP signal using $FRET_{Eff} = 1 - \frac{D_{pre}}{D_{post}}$, where D is the mean intensity of the donor CFP in the area where the acceptor was bleached, before (Dpre) and after (Dpost) acceptor bleaching. The FRET efficiency is considered positive when Dpost > Dpre. The image and statistical analyses were performed with the FRET module for the ZEN software (Zeiss). The average FRET efficiency and its standard deviation were calculated from the FRET efficiencies of each individual cell in 27–30 cells per experiment. The standard Student's $t$-test was used to determine the statistical significance of the results.

## HMTase reconstitution assay

In vitro HMTase reactions were modified from (Rea et al., 2000; Tachibana et al., 2001) as follows: 20 µl of reaction mixture containing 3.3 µM Histone 3.2 (NEB), 1 µM His-GST-KYP, and 50 nCi of S-adenosyl-[methyl-$^{14}$C]-L-methionine in HMTase buffer (50 mM Tris-HCl pH 9, 10 mM $MgCl_2$, 1 mM ß-mercaptoethanol, 250 mM sucrose) was incubated for 0–10 min at 37°C. The reaction products were separated by 18% SDS-polyacrylamide gel electrophoresis and visualized by Coomassie Brilliant Blue R-250 staining; then, the gels were fixed 1 hr in fixing solution (25% Ethanol, 2% Glycerol) and scintillated for 30 min in 1 M sodium salicylate. Gels were dried 2 hr at 80°C. The $^{14}$C signal was detected by fluorography using in a preflashed Classic autoradiography film blue sensitive; Filters Kodak Wratten No. 22 and No. 96 were used together for preflashing. Preflashed film was exposed 5–7 days at −80°C. The film was developed using a Kodak M35A X-OMAT Processor, and the results were digitalized in a Chemi-Doc XRS System and analyzed with the Image Lab Software (Bio-Rad, Hercules, CA, United States).

To assess the impact of TrAP on KYP activity, His-MBP-TrAP or His-MBP were pre-incubated with His-GST-KYP in different molar ratios, ranging from 0 to 10, for 1 hr at room temperature, then the assays were proceeded as described above. The experiments were performed 3–5 times for statistical analysis.

## ChIP assays

The analysis of histone modifications was performed as described (Saleh et al., 2008). Two grams of 9-day-old seedlings were crosslinked with 1% formaldehyde for 10 min by vacuum infiltration at 4°C; the reaction was stopped with 2 M Glycine to a final concentration of 100 mM at room temperature. Plants were rinsed 5 times with ice cold with water, flash-frozen in liquid nitrogen, and ground with mortar and pestle. The powder was suspended in 6 vol (12 ml) of nuclei isolation buffer (15 mM PIPES-KOH pH 6.8, 0.25 M sucrose, 0.9% Triton X-200, 5 mM $MgCl_2$, 60 mM KCl, 15 mM NaCl, 1 mM $CaCl_2$, 1 mM PMSF, 1 pellet/50 ml Complete EDTA-free Protease inhibitor [Roche]), filtered through two layers of Miracloth and centrifuged at 11,000 × rcf for 10 min in 4°C. After discarding the supernatant, the pellet was resuspended in 1 ml of Nuclei lysis buffer (50 mM HEPES pH 7.5, 1 mM EDTA pH 8.0, 150 mM NaCl, 1% SDS, 0.1% Sodium Deoxycholate, 1% Triton X-100, 1 pellet/50 ml Complete EDTA-free Protease inhibitor [Roche]); the samples were sonicated in 10 cycles 30 s ON and 90 s OFF, using

the Bioruptor (Diagenode, Belgium) at the highest power in 4°Cs. The sonicated samples were centrifuged for 10 min at 21,000 × rcf in 4°C. 100 µl of the clarified chromatin was diluted 10-fold with Nuclei lysis buffer without SDS for each assay. The immunoprecipitation was accomplished by the addition of 40 µl Protein A Dynabeads (Invitrogen) and 3 µl of the pertinent antibody, followed by 6 hr incubation at 4°C on mild rotation. The beads-conjugated complexes were washed with 1 ml of Low salt buffer (20 mM Tris-HCl pH 8.0, 2 mM EDTA pH 8.0, 150 mM NaCl, 0.5% Triton X-100, 0.2% SDS), followed by 1 ml of high salt buffer (20 mM Tris-HCl pH 8.0, 2 mM EDTA pH 8.0, 500 mM NaCl, 0.5% Triton X-100, 0.2% SDS), then with 1 ml of LiCl buffer (10 mM Tris-HCl pH 8.0, 1 mM EDTA pH 8.0, 250 mM LiCl, 1% sodium deoxycholate, 1% NP-40), and finally twice with 1 ml of TE (10 mM Tris-HCl, 1 mM EDTA, pH 8.0) by incubating 5 min at 4°C in between washes. The samples were eluted twice at room temperature with 250 µl of elution buffer (100 mM NaHCO₃, 0.5% SDS), for 15 and 30 min, respectively. The samples were decrosslinked with 100 mM NaCl at 65°C overnight, followed by Proteinase K treatment for 90 min at 45°C. The DNA was purified by Phenol:Chloroform: IsoamylAlcohol 25:24:1, and precipitated in 100% Ethanol at −80°C.

The antibodies used are mono-clonal anti-H3K9me2 (Abcam, #Ab1220); monoclonal anti-H3K4me3 (Millipore, cat #04-745); mono-clonal anti-H3K27me3 (Millipore, cat #07-449).

The immunoprecipitation of Flag-4Myc-KYP-Chromatin complexes was done as in (*Wierzbicki et al., 2008*), using Anti-FLAG M2 magnetic beads (Sigma-Aldrich, Cat# M8823). Two grams of rosette leaves 1–12 of mock or CaLCuV inoculated plants at 18 dpi were crosslinked with 1% formaldehyde for 25 min by vacuum infiltration at 4°C; the reaction was stopped with 2 M Glycine to a final concentration of 100 mM. Plants were rinsed five times with ice cold with water, flash-frozen in liquid nitrogen, and ground with mortar and pestle. The powder was suspended in 5 vol (10 ml) of Honda Buffer (20 mM HEPES-KOH pH 7.4, 0.44 M sucrose, 1.25% ficoll, 2.5% Dextran T40, 10 mM MgCl₂, 0.5% Triton X-100, 5 mM DTT, 2 mM PMSF, 1 pellet/25 ml Complete EDTA-free Protease inhibitor [Roche]), filtered through two layers of Miracloth and centrifuged at 2000 × rcf for 15 min in 4°C. After discarding the supernatant, the nuclear precipitates were washed three times with 1 ml of Honda buffer; subsequently, the pellet was suspended in 300 µl of Nuclei lysis buffer (50 mM Tris-HCl pH 8.0, 10 mM EDTA pH 8.0, 1% SDS, 2 mM PMSF, 1 pellet/25 ml Complete EDTA-free Protease inhibitor [Roche]) and sonicated in ten cycles 30 s ON and 90 s OFF, using the Bioruptor (Diagenode) at the highest power in 4°Cs. The sonicated samples were centrifuged for 10 min at 21000 × rcf in 4°C. 100 µl of the clarified chromatin was diluted 10-fold with ChIP dilution buffer (16.7 mM Tris-HCl pH 8.0, 1.2 mM EDTA pH 8.0, 167 mM NaCl, 1.1% Triton X-100, 1 pellet/25 ml Complete EDTA-free Protease inhibitor [Roche]) per ChIP. The immunoprecipitation was accomplished by the addition of 40 µl of Anti-FLAG M2 magnetic beads (Sigma-Aldrich, Cat# M8823), followed by 2 hr incubation at 4°C on mild rotation. The beads were washed five times with 1 ml of Washing buffer (20 mM Tris-HCl pH 8.0, 2 mM EDTA pH 8.0, 150 mM NaCl, 1% Triton X-100, 1% SDS, 2 mM PMSF, 1 pellet/25 ml Complete EDTA-free Protease inhibitor [Roche]) incubating 5 min at 4°C in between; then, two more washes with 1 ml TE buffer incubating 5 min at 4°C. Finally, the samples were eluted twice at room temperature with 125 µl of Elution buffer (100 mM NaHCO₃, 0.5% SDS), for 15 and 30 min, respectively. The samples were decrosslinked and the DNA extracted as above.

## Microarray analysis

Microarray analyses using the Affymetrix ATH1 platform were performed with two biological replicates using wild-type plants, *35S-TrAP* transgenic plants, and *lhp1* mutants. Seedlings were grown on MS medium with 1% sucrose for 7 days. One mg of total RNA was used for reverse transcription using MessageAmp II aRNA kits (Ambion) and 15 mg of labeled cRNA for hybridization. GeneChip hybridization and scanning were performed at the Genomics Resource Center, Rockefeller University, New York.

Statistical analysis of microarray data was performed using R software. Initially the microarray plates were tested for quality by an M plot and the data normalized by the RMA method from the Affy package. Subsequently, the distribution of the samples was assessed with scatter plots and the normalized data sets were evaluated with the Moderate t-test from R package limma for p-value computation. Then, the eBays method was used to compute moderated t-statistics and log-odds of differential expression by empirical Bayes shrinkage of the standard errors towards a common value.

The moderated t-statistic (t) is the logFC to its standard error. In our DEG results our thresholds are p-value < 0.05 and logFC >1 (up-regulated) or logFC < −1 (down-regulated). The False Discovery Rate was approximated from the eBayes adjusted p-value.

The significance of the overlapping data sets was calculated through Pearson's Chi-squared test with 1° of freedom.

## Quantitative PCR and RT-PCR

Expression levels of the tested genes were examined by quantitative RT-PCR. Total RNAs were prepared from 9 days-old seedlings and treated with DNase before being subjected to cDNA synthesis using Superscript III reverse transcriptase (Invitrogen) primed by random primers. The EF1a gene (*Williams et al., 2005*) was included as an internal control for normalization. The enrichment levels of specific genes after ChIP assay were also tested by quantitative PCR. Primers are listed in *Supplementary file 8*. The quantitative PCRs were performed in 384-well plates with an ABI7900HT real-time PCR system using the SYBR Green I master mix (Applied Biosystems, Waltham, MA, United States) in a volume of 10 µl. PCR conditions were as follows: 50°C for 2 min, 95°C for 10 min, 45 cycles of 96°C for 10 s followed by 60°C for 1 min. Three biological repeats were performed, and the reactions were performed in triplicate for each run. The comparative CT method was used to evaluate the relative quantities of each amplified product in the samples. The threshold cycle (CT) was automatically determined for each reaction by the system according to the default parameters. The specificity of the PCR was determined by dissociation curve analysis of the amplified products using the standard method installed in the system.

## Whole genome bisulfite sequencing

Approximately 500 ng of genomic DNA were used to generate libraries as described (*Feng et al., 2011*) using premethylated adapters (NEXTFlex Bisulfite-Seq Adapters #511911, Bioo Scientific, Austin, TX, United States). The adaptor-ligated fragments were purified by QIAQuick column (Qiagen) and bisulfite converted using the EpiTect Kit (Qiagen, Germany) following the manufacturer's instructions. The converted DNA was later enriched by 15 cycles of PCR using the Pfu Turbo Cx Polymerase (Agilent, Santa Clara, CA, United States), using the specific primers provided by Bios Scientific for enrichment. The library was finally purified with Agencourt AMPure XP beads (Beckman Coulter, Pasadena, CA, United States) according to manufacturer's instruction. The libraries were single-end sequenced using HiSeq High Output with read length of 50 bp. Base calling and sequence cleaning was performed with the standard Illumina software, then the clean reads were mapped to the *Arabidopsis* genome (Version: TAIR10) using Bismark v0.14.3 (*Krueger and Andrews, 2011*) with default parameters; the PCR duplicates were removed, and the methylation information was obtained with bismark with cutoff 3. The DMRs were obtained using swDMR (*Wang et al., 2015*) with window 200, step size 100, (left 1000, right 1000), the samples were compared using the Kruskal–Wallis analysis of variance with p-value < 0.01. The DMRs were then annotated using BEDtools (*Quinlan et al., 2010*).

## CaLCuV pathogenesis assays

3-week-old Col-0 wild-type, *kyp* mutant and *Flag-4Myc-KYP* complemented were infected by agroinfiltration of the CaLCuV infective clones of DNAA and DNAB pNSB1090 and pNSB1091 (*Arguello-Astorga et al., 2007*); the progression of the disease was evaluated daily in terms of time of symptom development and the severity of the symptoms observed. The assays were replicated 3 times, 36 plants of each genotype were used per assay, grown in short day conditions (8 hr light/16 hr dark).

To assess systemic infection, we conducted virus infection assays using CaLCuV or CaLCuV Δ*trap* on thirty wild-type and thirty *kyp* mutant plants grown in short day condition (16 hr dark/8 hr light) at eight true leaves developmental stage, then we harvested nine to eleven newly emerged rosette leaves of each individual 18 days after inoculation (*Figure 9B*).

## Acknowledgements

We thank Linda Hanley-Bowdoin, Judith Brown, and David Bisaro for the infective clones of the geminiviruses, Paul de Figueiredo and Hisashi Koiwa for imaging facilities, and Craig Kaplan for

the sonication instrument. We thank Charlie Johnson, Richard Metz, and Sabrina Moisan at Texas A&M Genomics and Bioinformatics Service for their excellent support for the bisulfite sequencing. We thank Joshua Yuan and Xin Wang for advice and technical support for the proteomics analysis. We thank Robert Klein as well as members of Zhang's laboratory for stimulating discussions and critical review of the manuscript. The work was supported by grants from the US National Science Foundation (NSF CAREER) (MCB-1253369), the US National Institutes of Health (R21AI097570) and the Welch foundation (A-1777) to XZ.

## Additional information

### Funding

| Funder | Grant reference | Author |
| --- | --- | --- |
| National Institutes of Health (NIH) | R21 AI097570 | Xiuren Zhang |
| Welch Foundation | A-1777 | Xiuren Zhang |
| National Science Foundation (NSF) | MCB-1253369 | Xiuren Zhang |

The funders had no role in study design, data collection and interpretation, or the decision to submit the work for publication.

### Author contributions

CC-G, XZ, Conception and design, Acquisition of data, Analysis and interpretation of data, Drafting or revising the article, Contributed unpublished essential data or reagents; XL, X-JW, Conception and design, Acquisition of data, Analysis and interpretation of data; CH, CZ, TH, FS, YZ, XZ, Acquisition of data, Analysis and interpretation of data, Contributed unpublished essential data or reagents; ZM, Acquisition of data, Analysis and interpretation of data, Drafting or revising the article

## Additional files

### Supplementary files

• Supplementary file 1. Microarray analysis of *35S-TrAP* transgenic plants.

• Supplementary file 2. Transcriptome comparison of the genes deregulated by *TrAP* overexpression and *lhp1* loss-of-function mutant.

• Supplementary file 3. Expression levels of genes encoding for TGS components in TrAP transgenic plants.

• Supplementary file 4. List of proteins tested for interaction with TrAP.

• Supplementary file 5. CG methylation analysis of Col-0 wild type, *kyp* mutant, and TrAP transgenic plants.

• Supplementary file 6. CHG methylation analysis of Col-0 wild type, *kyp* mutant, and TrAP transgenic plants.

• Supplementary file 7. CHH methylation analysis of Col-0 wild type, *kyp* mutant, and TrAP transgenic plants.

• Supplementary file 8. List of primers used in this article.

## Major datasets

The following datasets were generated:

| Author(s) | Year | Dataset title | Dataset ID and/or URL | Database, license, and accessibility information |
|---|---|---|---|---|
| Castillo-González C, Zhang X, Liu X, Wang X | 2015 | TrAP expression in Arabidopsis thaliana lead to global reprogramming of endogenous genes | http://www.ncbi.nlm.nih.gov/geo/query/acc.cgi?acc=GSE73495 | Publicly available at the NCBI Gene Expression Omnibus (Accession no: GSE73495). |
| Castillo-González C, Zhang X, Liu X, Wang X | 2015 | Geminivirus-encoded TrAP suppressor inhibits the histone methyltransferase SUVH4/KYP to counter host defense | http://www.ncbi.nlm.nih.gov/geo/query/acc.cgi?acc=GSE73586 | Publicly available at the NCBI Gene Expression Omnibus (Accession no: GSE73586). |

The following previously published dataset was used:

| Author(s) | Year | Dataset title | Dataset ID and/or URL | Database, license, and accessibility information |
|---|---|---|---|---|
| Du J, Johnson LM, Groth M, Feng S, Hale CJ, Li S, Vashisht AA, Gallego-Bartolome J, Wohlschlegel JA, Patel DJ, Jacobsen SE | 2014 | Crystal structure of KRYPTONITE in complex with mCHH DNA and SAH | http://www.rcsb.org/pdb/explore/explore.do?structureId=4QEN | Publicly available at RCSB Protein Data Bank (Accession no. 4QEN). |

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
