## [Decision Letter]

Thank you for sending your work entitled “Geminivirus-encoded TrAP suppressor inhibits SUVH4/KYP activity to counter host defense” for consideration at *eLife*. Your article has been favorably evaluated by Detlef Weigel (Senior Editor) and three reviewers. All four discussed their comments before we reached this decision, and the editor has assembled the following comments to help you prepare a revised submission.

In your manuscript, you propose that a geminivirus encoded protein, TrAP, acts by interfering with the action of the SET domain histone methyltransferase SUVH4, a component of the plant host's transcriptional gene silencing machinery, providing a means that promotes virus survival in the arms race between virus and host. The topic is of broad interest given the conservation of HMTases in multicellular eukaryotes and the existence of other DNA viruses that affect animals, including humans. Your findings expand upon the previously proposed pleiotropic functions of TrAP proteins in suppression of TGS, such as adenosine kinase inhibition and SAMDC1 stabilisation. As well as illustrating a new role for TrAP during viral pathogenesis, your work could provide a useful tool for scientists, who could employ TrAP as a dominant inhibitor of H3K9me modification.

Your conclusions regarding the TrAP/SUNVH4 interaction are based on *TrAP* overexpression causing developmental defects reminiscent of certain epigenetic mutants, on physical interaction of TrAP with SUVH4 and inhibition of SUVH4 histone methyltransferase activity in vitro, and on SUVH4 effects on the viral genome. While there was in principle broad enthusiasm for the work among the reviewers, there were several substantial concerns that preclude acceptance of the work in its current form. However, we would be very interested in a substantially revised version, in which you provide:

1) Whole-genome comparisons of DNA and H3K9 methylation in *suvh4* mutants and *TrAP* overexpressers.

2) More evidence of direct interaction of SUVH4 and TrAP using co-IPs from complemented mutants.

The first major point of your manuscript concerns the interaction of TrAP with SUVH4. While we were convinced that TrAP can interact with SUVH4 in vitro, the in vivo data with overexpression constructs were less convincing. This is especially important considering there might be competition for binding to other SET domain proteins. As is, it is difficult to determine whether TrAP is indeed exerting its effect largely through SUVH4 rather than through any of the other 31+ *Arabidopsis* SET domain proteins or any of the other previously documented TrAP targets. The differences in phenotypes between *TrAP* overexpressers and *suvh4* mutants would suggest that the SUVH4 interaction is not the entire story. To further support the conclusion that TrAP specifically interacts with SUVH4 in vivo, you should conduct co-IP experiments using constructs expressed at endogenous levels. You already have a tagged version of SUVH4 expressed under its endogenous promoter and you have many plant lines expressing varying levels of TrAP, and these lines could be easily crossed. Even better, but not mandatory, would be affinity purification of the epitope tagged TrAP followed by mass spectrometry.

A second main point is that TrAP inhibits SUVH4. Again, the in vitro data are very good, but the in vivo data are less conclusive. For example, you demonstrate that constitutive expression of TrAP causes developmental defects reminiscent of mutations affecting various epigenetic modifications. However, such defects are not observed in the *suvh4* single mutant, nor the *suvh456* triple mutant, in which H3K9me levels are drastically reduced. In fact, you start out by comparing *TrAP* overexpressers with *lhp1* mutants. In any case, the phenotypic discrepancies suggests that expression of TrAP disrupts many biological processes not connected with SUVH4 or its closest homologs, and it raises the possibility that TrAP generally targets SET domain HMTases, a possibility that you acknowledge in the Discussion. In and of itself, such potential promiscuity of TrAP is not an issue, as it would be just as interesting if a single viral protein could inhibit many SET domain HMTases. However, to make the argument that part of the effects of TrAP are through inhibition of SUVH4, you should compare the effects of *TrAP* overexpression to the effects of *suvh4* mutants rather than the effects of other chromatin regulatory factors like LHP1. As there are very few genes mis-regulated in *suvh4* single mutants, you should compare global levels of DNA and H3K9 methylation in *TrAP* overexpressers to those observed in a *suvh4* mutant (or *suv456* triple mutants). We also note that LHP1 has been shown to bind H3K27me3 and that it is in the CURLY LEAF/Polycomb pathway, and that there is no evidence that LHP1 is related to SUVH4 function. It is not clear how these distinct effects can be easily reconciled, and it tends to make the current presentation appear disjointed. On the other hand, in Figure 5 you show that *TrAP* overexpression tends to phenocopy methylation patterns of *suvh4* mutants at several endogenous loci, more so than the *lhp1* knockout. These issues would most easily be reconciled with whole-genome data, as suggested above.

A final main point concerns the hypothesis that TrAP-mediated inhibition of SUVH4 is a means of suppressing host defense. In support of this notion, you cite (and reproduce) previous data showing that *suvh4* mutants are more susceptible to geminivirus infection. You also show that SUVH4 associates with the geminivirus genome by ChIP. However H3K9me modification of the viral genome is largely unaffected by loss of SUVH4, and there is no evidence that the observed changes influence expression of viral genes. You should compare viral gene expression in wild type and *suvh4* mutants, to determine if transcriptional gene silencing by SUVH4 does indeed affect viral gene expression. If no change in expression is observed in the *suvh4* mutant, this would argue against the presented model and suggest TrAP is functioning via an alternative mechanism.

[Editors' note: further revisions were requested prior to acceptance, as described below.]

Thank you for resubmitting your work entitled “Geminivirus-encoded TrAP suppressor inhibits the histone methyltransferase SUVH4/KYP to counter host defense” for further consideration at *eLife*. Your revised article has been favorably evaluated by Detlef Weigel (Senior Editor) and three reviewers. The manuscript has been improved but there are some remaining issues that need to be addressed before acceptance, as outlined below.

Specifically, we would like you to follow up as much as possible the suggestions of Reviewer 2, and also to provide BS-seq data.

Reviewer #1:

The authors have addressed all my concerns. I understand the difficulties the authors had to address the major concern #2 and I think that, even when the requested experiment would improve the manuscript its absence doesn't make the manuscript any weaker.

Reviewer #2:

In the revised version of their manuscript, Castillo-Gonzalez et al. addressed several, but not all, of my main concerns. Regarding the in vivo relevance of the KYP-TrAP interaction, the authors conducted TrAP IP mass spectrometry experiments and found many peptides corresponding to KYP, confirming an interaction under biologically relevant conditions. A second concern was in regards to the H3K9me2 ChIP data supporting the conclusion that *TrAP* overexpression affects the activity of KYP in vivo. As outlined below, I remain concerned about the interpretation of this data. Regarding the final point, the role of KYP in regulating the expression of virus genes, the authors provide additional genetic evidence supporting a role of KYP in virus infection in a manner that is specifically connected to the *TrAP* gene. While this is not a direct assessment of the role of KYP in transcriptional regulation of viral genes, it does further support a link between TrAP, KYP, and viral defense. Taken together, the new experiments presented significantly strengthen the connection between KYP and TrAP, but the extent to which the TrAP protein modulates KYP HMTase activity at endogenous loci remains less clear.

ChIP data:

Castillo-Gonzalez present H3K9me ChIP data at a total of 14 endogenous loci in Figure 5 and conclude that ∼11/14 show reduced H3K9me levels in the *TrAP* overexpression lines which is in support of their hypothesis that TrAP inhibits KYP activity in vivo. However, as detailed below, it is not clear which, if any of the 10 genes tested in Figure 5 are normally marked H3K9me2, as would be expected for bona fide KYP targets. Without further clarification on these loci, that leaves the 4 transposons tested in Figure 5, of which only two show a >2x reduction in signal in the *TrAP* overexpression lines and the *kyp* mutants. Thus, while this data is in support of their conclusion that TrAP affects KYP activity in vivo, it is unclear how broadly TrAP is affecting KYP function. Inclusion of more loci known to be targets of KYP would significantly strengthen the conclusion that TrAP affects KYP function in vivo.

The ChIP data presented in Figure 5/B should be separated out to indicate which genes are marked by each chromatin modification in WT plants. Each gene will be predominantly marked by only one modification (H3K4me, H3K9me, or H3K9me). Without knowing the levels of each modification in WT plants, the data cannot be fully interpreted as presented because the WT sample is artificially set to one. For example if a gene (like RDR1) is expressed and marked by H3K4me in Wt plants, but is not marked by H3K9me or H3K27me then a ChIP signal showing reduced H3K9me or H3K27me is likely noise rather than a real signal. This is a critical point in the paper, as they use this data make the statement that TrAP causes decreased H3K9me levels. In the subsection “TrAP reduced H3K9me2 repressive marks in vivo” they state: “specifically, eight out of 10 TrAP deregulated loci, including the flowering-promoting genes, displayed reduced H3K9me in the *TrAP* overexpression plants compared to WT plants…”. However, in looking at available H3K9me2 data on the UCSC browser, these genes are largely devoid of H3K9me signal. To determine if there is H3K9me2 signal at the sites tested in Figure 5 in WT plants, and to determine if this signal is meaningfully reduced in the *TrAP* overexpression lines, the WT signal should not be normalized to one and instead should be presented as a % input (and/or normalized to H3) alongside the enrichment level at a gene without H3K9me (like actin). Such analysis should not require repeating the ChIP assays and thus can be done in the absence of a good lot of Abcam 1220 antibody. However, if these sites do not have sufficient H3K9me2 signal in Wt plants then testing additional loci will become necessary. In this case, looking at DNA methylation levels (even if they have been shown to be affected by other TrAP proteins the Bisaro lab) in Wt plants, *TrAP* overexpressor lines, and *kyp* mutants can serve in support of the authors’ claims without relying on the Abcam H3K9me2 antibody.

Mass spectrometry data:

The text referring to the TrAP mass spectrometry data is quite brief and ambiguous as written. For example it is stated that 31 peptides uniquely match KYP, but it is unclear if these were only recovered in induced samples. Additional details should be provided in the text. For example in the response to the reviewers it is clearly stated that no peptides to KYP were identified in Col controls and that KYP was the most abundant hit from the mass spectrometry. It is not clear why these important details would be omitted from the text.

Reviewer #3:

The manuscript has improved, and I can understand the difficulties with the H3K9me2 ChiP-seq, but I dont really agree with the reasoning to not do the BSseq. This is easy enough and while changes in DNA methylation might represent a downstream effect, the DNA methylation changes are well characterised in in suvh4 so they could still get a good idea of whether or not *TrAP* overexpression mimics loss of SUHV4 activity. The papers that the authors mention “Dr. David Bisaro's lab (Buchmann et al., Figure 5 in 2009 JV. 83 p5005-5013; and Raja, et al., Figure 5 in 2008, JV. 82 p8997-9007)” only perform BSseq at specific loci, not genome wide.

---

## [Author Response]

*The first major point of your manuscript concerns the interaction of TrAP with SUVH4. While we were convinced that TrAP can interact with SUVH4* in vitro*, the* in vivo *data with overexpression constructs were less convincing. This is especially important considering there might be competition for binding to other SET domain proteins. As is, it is difficult to determine whether TrAP is indeed exerting its effect largely through SUVH4 rather than through any of the other 31+* Arabidopsis *SET domain proteins or any of the other previously documented TrAP targets. The differences in phenotypes between* TrAP *overexpressers and* suvh4 *mutants would suggest that the SUVH4 interaction is not the entire story. To further support the conclusion that TrAP specifically interacts with SUVH4* in vivo*, you should conduct co-IP experiments using constructs expressed at endogenous levels. You already have a tagged version of SUVH4 expressed under its endogenous promoter and you have many plant lines expressing varying levels of TrAP, and these lines could be easily crossed. Even better, but not mandatory, would be affinity purification of the epitope tagged TrAP followed by mass spectrometry*.

Thanks for this point. To further confirm the interaction between SUVH4/TrAP, we have pursued proteomics analysis of the TrAP protein complexes by mass spectrometry. To this end we used stable *Arabidopsis thaliana* transgenic lines carrying the *XVE::Flag-Myc4-TrAP* construct. The XVE promoter is inducible with β-estradiol and allows for time-regulated expression of the viral protein. For this experiment we used 9-days old seedlings and we induced TrAP expression for 16h, then we purified the TrAP complexes through two-step affinity purification to study whether we could recover endogenous KYP protein, we performed this experiment in parallel with Col-0 plants. The mass spectrometry analysis recovered 31 unique peptides that matched specifically to KYP/SUVH4, none of which were found in the Col-0 control plants. Moreover, KYP was among the most abundant recovered proteins in the immunoprecipitated TrAP complexes. We have added these results in the new Figure 3—figure supplement 1. Thus, we are now very confident that KYP is a bona fide target of TrAP in vivo.

Importantly, we proposed in our manuscript that TrAP targets KYP to counter host defense. However, we did not mean that KYP is the only target of TrAP in the host. Indeed, we found by in vitro pull down assay that TrAP also interacts with the KYP paralogs SUVH2, 5, and 6 (Figure 4—figure supplement 2), and we also discussed the possibility of a more promiscuous effect of TrAP in the SET domain proteins predicted from the morphological phenotypes (in the Discussion). In fact, one aspect of our future effort is to examine how many SET domain containing proteins does TrAP target in the host, and how many of those targets are involved in host defense mechanisms. Also, we are working on additional potential targets recovered by proteomic analysis in our future study on this project.

*A second main point is that TrAP inhibits SUVH4. Again, the* in vitro *data are very good, but the* in vivo *data are less conclusive. For example, you demonstrate that constitutive expression of TrAP causes developmental defects reminiscent of mutations affecting various epigenetic modifications. However, such defects are not observed in the* suvh4 *single mutant, nor the* suvh456 *triple mutant, in which H3K9me levels are drastically reduced. In fact, you start out by comparing* TrAP *overexpressers with* lhp1 *mutants. In any case, the phenotypic discrepancies suggests that expression of TrAP disrupts many biological processes not connected with SUVH4 or its closest homologs, and it raises the possibility that TrAP generally targets SET domain HMTases, a possibility that you acknowledge in the Discussion. In and of itself, such potential promiscuity of TrAP is not an issue, as it would be just as interesting if a single viral protein could inhibit many SET domain HMTases. However, to make the argument that part of the effects of TrAP are through inhibition of SUVH4, you should compare the effects of* TrAP *overexpression to the effects of* suvh4 *mutants rather than the effects of other chromatin regulatory factors like LHP1. As there are very few genes mis-regulated in* suvh4 *single mutants, you should compare global levels of DNA and H3K9 methylation in* TrAP *overexpressers to those observed in a* suvh4 *mutant (or* suv456 *triple mutants). We also note that LHP1 has been shown to bind H3K27me3 and that it is in the CURLY LEAF/Polycomb pathway, and that there is no evidence that LHP1 is related to SUVH4 function. It is not clear how these distinct effects can be easily reconciled, and it tends to make the current presentation appear disjointed. On the other hand, in*
Figure 5
*you show that* TrAP *overexpression tends to phenocopy methylation patterns of* suvh4 *mutants at several endogenous loci, more so than the* lhp1 *knockout. These issues would most easily be reconciled with whole-genome data, as suggested above*.

This is an insightful point. It is true that TrAP transgenic plants phenocopy the morphology of mutants in the lhp1/leafy/polycomb pathway, but they also recapture the molecular phenotype of *suvh4*. We do not have to reconcile the relevance of LHP1 with SUVH4 (in fact, they are independent pathways in plants) since TrAP could target both pathways as stated in the discussion. We agree that our manuscript would be further improved if we could conduct whole-genome comparisons of H3K9 methylation in *suvh4* mutants and TrAP transgenic lines. With the advice as a guide, we have been actively pursuing this experiment. We prepared all materials and were ready for the ChIP-seq. However, we experienced an unexpected problem with the antibodies: as you know, the Abcam anti-H3K9me2 antibody (Ab1220) is the preferred antibody for ChIP-seq, and it is also the one we have successfully used before. Unfortunately, we ran out of our previous antibody (lot number 138716-4), and the newly purchased lots have not worked. We have screened the antibody from three lots already, with only one replacement barely immunoprecipitating H3K9me2 (signal 10^3^ fold less than the previous antibody). To date, we have spent many weeks doing all kinds of tests in all possible conditions. The conclusion is that we need to wait for a new lot of antibody from Abcam to be tested, and to perform the ChIP-seq if it is good.

I thought to mention that the reason for us to choose the Abcam antibody was that Abcam is known to produce good anti-H3K9m2 antibody. In a big collaborative project funded by NIH to screen varieties of antibodies from many vendors worldwide, to filter out noise and to find the best ones in the epigenetics field (Egelhofer, et al., 2011. An assessment of histone-modification antibody quality. 18:91-93), Abcam’s ab1220 was reported as the best antibody for H3K9me2 ChIP. A website was provided for evaluation of the performance of each antibody (i.e. http://compbio.med.harvard.edu/antibodies/targets/37 and

http://compbio.med.harvard.edu/antibodies). There are several other companies producing anti-H3K9me2 (i.e. Active motif; Diagenode; Hiroshi Kimura Lab; and millipore). Data from this project suggested that the antibodies from these vendors could not be used for ChIP-seq.

We understand the concerns brought up by the reviewers, and we agree that this experiment can improve our manuscript. Nonetheless, in order for us to present solid work to the scientific community, we require high-quality antibody for the ChIP-seq assay. While we could continue to test the new antibodies from different lots from Abcam and other sources, there is no guarantee of when and where we could obtain the high-quality anti-H3K9me2 antibody. However, to clarify the concern from this reviewer, we pursued an alternative genetic approach: First of all, we performed Southern blot analysis of virus titers in *kyp* mutant and Col-0 (new Figure 8), and observed substantial increase of virus replication in the *kyp* mutant compared to wild-type plants. This result indicates that KYP is a major immune system for defense of virus. More importantly, we created a mutant virus without functional TrAP (CaLcV ∆TrAP), inoculated *kyp* mutants and Col-0 control. We found that control plants did not allow replication of CaLCuV ∆TrAP at a detectable level. In contrast, the *kyp* mutant plants displayed systemic virus accumulation to a certain extent (as shown in single-stranded viral DNA, New Figure 8). Thus, these new results clearly boosted our model that TrAP targets KYP to counter host defense (also see response below).

Given our extensive genetic and biochemical evidence in the manuscript and new proteomics data, we believe that TrAP functionally targets KYP to counter host defense. At this stage, we would rather leave whole-genome comparisons of H3K9 methylation in *suvh4* mutants and *TrAP* transgenic lines for a follow-up work. Regarding comparison of DNA methylation levels in *suvh4* and *TrAP* transgenic lines, we would like to mention that this assay is not a priority for this manuscript – the reason is that the impact of TrAP and SUVH4 on host and viral genomes has been reported by Dr. David Bisaro's lab (Buchmann et al., Figure 5 in 2009 JV. 83 p5005-5013; and Raja, et al., Figure 5 in 2008 JV. 82 p8997-9007). These results are well in line with our current work. Thus, additional sequencing might not increase the novelty and concept advance of our study; moreover, our biochemical study shows that TrAP mainly interacts with SET domain of KYP, which is the methyltransferase domain, and we presume that changes in histone methylation might be the primary cause for the TrAP function, while DNA methylation changes might be a downstream effect.

*A final main point concerns the hypothesis that TrAP-mediated inhibition of SUVH4 is a means of suppressing host defense. In support of this notion, you cite (and reproduce) previous data showing that* suvh4 *mutants are more susceptible to geminivirus infection. You also show that SUVH4 associates with the geminivirus genome by ChIP. However H3K9me modification of the viral genome is largely unaffected by loss of SUVH4, and there is no evidence that the observed changes influence expression of viral genes. You should compare viral gene expression in wild type and* suvh4 *mutants, to determine if transcriptional gene silencing by SUVH4 does indeed affect viral gene expression. If no change in expression is observed in the* suvh4 *mutant, this would argue against the presented model and suggest TrAP is functioning via an alternative mechanism*.

Another insightful point. So far, we have provided evidence of the effect of KYP levels on the progression of CaLCuV infection in *Arabidopsis*, as the *kyp* mutants are hypersensitive, the complemented lines are as sensitive as the wild type, and the over expression lines are more resistant to infection. Moreover, we have shown that KYP expression affects the methylation status of the viral chromatin. Together, these data point to a direct effect of KYP in the pathogenesis of CaLCuV.

With the comment as a guidance, we have performed Southern blots to determine the populations of the viral DNA in the infected plants, and we have now shown that *kyp* mutant plants not only show more severe symptoms of infection, but also they accumulate the virus titers at more than 20-fold higher level than that in the wild type plants. The virus accumulation and systemic infection entails TrAP protein, as it is required for the expression of the late genes (coat protein and nuclear shuttle protein) that are necessary for the accumulation of the infective ssDNA particle. Most importantly, we have engineered a TrAP null mutant virus and we have performed infection studies in both the wild type and *kyp* mutant plants. As previously shown, begomoviruses deficient in TrAP are unable to cause systemic infection in wild type hosts; however, *kyp* mutant hosts can sustain systemic infection of this virus, although at low levels (Please refer to our response above).

[Editors' note: further revisions were requested prior to acceptance, as described below.]

*Specifically, we would like you to follow up as much as possible the suggestions of Reviewer 2, and also to provide BS-seq data*.

Reviewer #1:

*The authors have addressed all my concerns. I understand the difficulties the authors had to address the major concern #2 and I think that, even when the requested experiment would improve the manuscript its absence doesn't make the manuscript any weaker*.

Reviewer #2:

*In the revised version of their manuscript, Castillo-Gonzalez et al. addressed several, but not all, of my main concerns. Regarding the* in vivo *relevance of the KYP-TrAP interaction, the authors conducted TrAP IP mass spectrometry experiments and found many peptides corresponding to KYP, confirming an interaction under biologically relevant conditions. A second concern was in regards to the H3K9me2 ChIP data supporting the conclusion that* TrAP *overexpression affects the activity of KYP* in vivo*. As outlined below, I remain concerned about the interpretation of this data. Regarding the final point, the role of KYP in regulating the expression of virus genes, the authors provide additional genetic evidence supporting a role of KYP in virus infection in a manner that is specifically connected to the* TrAP *gene. While this is not a direct assessment of the role of KYP in transcriptional regulation of viral genes, it does further support a link between TrAP, KYP, and viral defense. Taken together, the new experiments presented significantly strengthen the connection between KYP and TrAP, but the extent to which the TrAP protein modulates KYP HMTase activity at endogenous loci remains less clear*.

ChIP data:

*Castillo-Gonzalez present H3K9me ChIP data at a total of 14 endogenous loci in*
Figure 5
*and conclude that ∼11/14 show reduced H3K9me levels in the* TrAP *overexpression lines which is in support of their hypothesis that TrAP inhibits KYP activity* in vivo*. However, as detailed below, it is not clear which, if any of the 10 genes tested in*
Figure 5
*are normally marked H3K9me2, as would be expected for bona fide KYP targets. Without further clarification on these loci, that leaves the 4 transposons tested in*
Figure 5*, of which only two show a >2x reduction in signal in the* TrAP *overexpression lines and the* kyp *mutants. Thus, while this data is in support of their conclusion that TrAP affects KYP activity* in vivo*, it is unclear how broadly TrAP is affecting KYP function. Inclusion of more loci known to be targets of KYP would significantly strengthen the conclusion that TrAP affects KYP function* in vivo.

This is a very important point. Obviously, we missed the opportunity of presenting our point clearly in the previous versions. Here in this version, we emphasized this critical point.

Briefly, we have included five transposable elements known to be targets of KYP, all of which showed greater than 2-fold decrease in our experiments (new Figure 6). On the other hand, the comprehensive body of work published by Dr. Steven Jacobsen group on chromatin structure and regulation (Zhang et al., Genome-wide analysis of mono-, di- and trimethylation of histone H3 lysine 4 in *Arabidopsis thaliana*, Genome Biology, 2009; Bernatavichute et al., Genome-Wide Association of Histone H3 Lysine Nine Methylation with CHG DNA Methylation in *Arabidopsis thaliana*, PLoS ONE, 2008; Zhang et al., PLoS Biology, 2007) allowed us to pinpoint the specific histone modifications (H3K9me2, H3K27me3 and H3K4me3) on the genome and to provide a guide for selection of the regions for ChIP-qPCR (see Figure 10).

Author response image 1.**DOI:**
http://dx.doi.org/10.7554/eLife.06671.031

In fact, the criteria we used to select the loci for RT-qPCR and ChIP-qPCR were: 1) their transcription was significantly affected by the expression of the TrAP transgene (Figure 2); and 2) H3K9me2 mark have been found in the loci. For example, FTM1, FT, AGL9, MFT, PRE1 and AT4G17810 all harbor H3K9me2 marks (Figure 10).

We should also mention that there might be few cases in which our data was not fully consistent with the published work. The reason is that plant materials in our experiments differ from the materials used in previous studies. We used 11 days-old seedlings grown in short day conditions (8h light/ 16h dark), while some of the data reported was obtained from 3 weeks old adult plants grown under constant light (i.e. H3K9me2 and H3K4me3), or from 10-14 days-old seedlings grown in long day condition (16h light/ 8h dark) (i.e. H3K27me3). These differences imply a huge disparity in the developmental stage of the samples, as H3K9me2 and H3K4me3 mark were measured from flowered adult plants. Taking into consideration that we focused on flowering genes, it is not surprising that some differences between our data and others’ exist.

*The ChIP data presented in*
Figure 5*/B should be separated out to indicate which genes are marked by each chromatin modification in WT plants. Each gene will be predominantly marked by only one modification (H3K4me, H3K9me, or H3K9me). Without knowing the levels of each modification in WT plants, the data cannot be fully interpreted as presented because the WT sample is artificially set to one. For example if a gene (like RDR1) is expressed and marked by H3K4me in Wt plants, but is not marked by H3K9me or H3K27me then a ChIP signal showing reduced H3K9me or H3K27me is likely noise rather than a real signal. This is a critical point in the paper, as they use this data make the statement that TrAP causes decreased H3K9me levels. In the subsection “TrAP reduced H3K9me2 repressive marks* in vivo*” they state: “specifically, eight out of 10 TrAP deregulated loci, including the flowering-promoting genes, displayed reduced H3K9me in the* TrAP *overexpression plants compared to WT plants…”. However, in looking at available H3K9me2 data on the UCSC browser, these genes are largely devoid of H3K9me signal. To determine if there is H3K9me2 signal at the sites tested in*
Figure 5
*in WT plants, and to determine if this signal is meaningfully reduced in the* TrAP *overexpression lines, the WT signal should not be normalized to one and instead should be presented as a % input (and/or normalized to H3) alongside the enrichment level at a gene without H3K9me (like actin). Such analysis should not require repeating the ChIP assays and thus can be done in the absence of a good lot of Abcam 1220 antibody. However, if these sites do not have sufficient H3K9me2 signal in Wt plants then testing additional loci will become necessary. In this case, looking at DNA methylation levels (even if they have been shown to be affected by other TrAP proteins the Bisaro lab) in Wt plants,* TrAP *overexpressor lines, and* kyp *mutants can serve in support of the authors’ claims without relying on the Abcam H3K9me2 antibody*.

This is another insightful point; and thanks to this reviewer for guiding us on how to present our ChIP data in a more accurate way.

First, based on the advice from this reviewer, we recalculated our ChIP-qPCR and completely reformatted the data taking into account the background signal, by redoing qPCR using the remaining ChIP DNA from the original experiments. In the new figures, each histone modification is shown separately. Also, the data is presented as enrichment relative to tubulin (TUB8) or relative to input alongside to TUB8 (this is particular for H3K9me2, since the TUB8 signal was extremely low and the values relative to TUB8 resulted in an exorbitant range). We would like to emphasize that our remaining ChIP-DNA from previous experiments only allowed us to repeat 7 of 12 protein-coding genes in our original manuscript, and the 5 transposable elements presented in the current version.

We would like to reiterate that the previously published H3K9me2 ChIP-seq data are obtained using plants at a developmental stage different from ours as stated above, and this fact could cause differences at the chromatin level that may render the data not readily comparable. For instance, the reported H3K9me2 marks at some of our target genes are low in the 3-week old plants, but one could not imply that they do not exist at the seedling stage. As an additional case, the published H3K27me3 (obtained from seedlings) and H3K4me3 (obtained from adult plants) available datasets, if directly compared, might mislead us to conclude that both modifications coexist at the flowering genes.

*Mass spectrometry data*:

*The text referring to the TrAP mass spectrometry data is quite brief and ambiguous as written. For example it is stated that 31 peptides uniquely match KYP, but it is unclear if these were only recovered in induced samples. Additional details should be provided in the text. For example in the response to the reviewers it is clearly stated that no peptides to KYP were identified in Col controls and that KYP was the most abundant hit from the mass spectrometry. It is not clear why these important details would be omitted from the text*.

We stated this point in the previous point-to-point response letter; and now we have moved this critical information in the main text.

Reviewer #3:

*The manuscript has improved, and I can understand the difficulties with the H3K9me2 ChiP-seq, but I dont really agree with the reasoning to not do the BSseq. This is easy enough and while changes in DNA methylation might represent a downstream effect, the DNA methylation changes are well characterised in in suvh4 so they could still get a good idea of whether or not* TrAP *overexpression mimics loss of SUHV4 activity. The papers that the authors mention “Dr. David Bisaro's lab (Buchmann et al.,*
Figure 5
*in 2009 JV. 83 p5005-5013; and Raja, et al.,*
Figure 5
*in 2008, JV. 82 p8997-9007)” only perform BSseq at specific loci, not genome wide*.

As requested, we have performed whole genome bisulfite sequencing in our Col-0 wild type, *suvh4/kyp*, and 35S-TrAP plants. The new result is now included in the new subsection “TrAP decreases CHH DNA methylation” and new Figure 7, Figure 7—figure supplement 1, and the [Supplementary-material SD5-data SD6-data SD7-data].

We found that *TrAP* overexpression did not affect CHG methylation; however, it reduced CHH methylation in gene rich regions that were almost completely overlapped with the *suvh4* mutant. These regions included FT, CUC1, and AT4G17810, in accordance with our reported loss of H3K9me2 enrichment in TrAP transgenic lines and *suvh4* mutants. Our new result is a bit surprising, but more exciting, that is to say, CHH and CHG could be regulated separately – it is known that CMT2 – the main CHH DNA methyltransferase – has greater affinity for H3K9me2 over H3K9me1. Other non-CG methyltransferases, such as CMT3 (which catalyzes CHG methylation) do not show binding preference among different methylation levels in the histone tail. We therefore envision methylation at CHH sites to be more sensitive to changes in H3K9me2 methylation than CHG. In all, our data is consistent with decrease in H3K9me2, which could be explained by partial inhibition of SUVH4 activity.